# BEAT: Tokenizing and Generating Symbolic Music by Uniform Temporal Steps

**Lekai Qian** [1]  **Haoyu Gu** [1]  **Jingwei Zhao** [2]  **Ziyu Wang** [3] [4]

## Abstract

Tokenizing music to fit the general framework of language models is a compelling challenge, especially considering the diverse symbolic structures in which music can be represented (e.g., sequences, grids, and graphs). To date, most approaches tokenize symbolic music as sequences of musical events, such as onsets, pitches, time shifts, or compound note events. This strategy is intuitive and has proven effective in Transformer-based models, but it treats the regularity of musical time implicitly: individual tokens may span different durations, resulting in non-uniform time progression. In this paper, we instead consider whether an alternative tokenization is possible, where a uniform-length musical step (e.g., a beat) serves as the basic unit. Specifically, we encode all events within a single time step at the same pitch as one token, and group tokens explicitly by time step, which resembles a sparse encoding of a piano-roll representation. We evaluate the proposed tokenization on music continuation and accompaniment generation tasks, comparing it with mainstream event-based methods. Results show improved musical quality and structural coherence, while additional analyses confirm higher efficiency and more effective capture of long-range patterns with the proposed tokenization.

## 1. Introduction

As language models continue to advance, it is increasingly compelling to explore how symbolic music generation can be integrated into this paradigm. A central challenge lies in the tokenization of music, as music is a highly structured form of information that exhibits regular temporal patterns, polyphonic organization, and rich expressive variation.

Music can be encoded in diverse ways, each highlighting different structural aspects: **grid-based representations**, such as piano-rolls, which discretize music into fixed time units (e.g., beats or tatums) and naturally exhibit temporal shift invariance; **notation-based representations**, such as ABC (Walshaw, 2011) or MusicXML (Good, 2001), which capture syntactic and hierarchical relationships; and **event-based representations**, such as MIDI (MIDI Manufacturers Association, 1996), which are well suited for representing temporally ordered performance actions.

Among these representations, event-based tokenizations have become dominant in symbolic music generation (Huang & Yang, 2020; Hsiao et al., 2021). Their one-dimensional, sequential structure provides a straightforward way to tokenize musical control messages and integrates naturally with Transformer-based language models. Recently, notation-based formats have also been explored (Wang et al., 2025). Both approaches represent music as chronologically ordered sequences of events; however, they do not explicitly encode the uniformity of the temporal grid, a property that is inherent to grid-based representations.

In event- or notation-based representations, each token, or the interval between successive tokens, may span a variable and uncertain duration, ranging from a fraction of a beat to multiple beats. As a result, the model must implicitly infer the underlying temporal grid, placing an additional burden on learning regular temporal structure. In contrast, human musical perception is strongly grounded in *evenly spaced temporal units*, such as beats or pulses, which serve as fundamental primitives to form higher-level music understanding (London, 2012; Huron, 2006). Motivated by this observation, we treat uniform temporal discretization as a core inductive bias in tokenization design and investigate how such representations can be naturally integrated with Transformer-based models.

In this paper, we instantiate the idea of grid-based tokenization as **BEAT** (**B**eat-wise **E**ncoding for **A**utoregressive **T**ransformers).[1] BEAT assumes a quarter note, referred to as a beat, as a fixed temporal unit. We first encode the information for each beat and each pitch into a single token. Tokens corresponding to all-rests are omitted, and the remaining

[1]South China University of Technology [2]National University of Singapore [3]Mohamed bin Zayed University of Artificial Intelligence [4]New York University. Correspondence to: Lekai Qian <lekaiqian@outlook.com>, Ziyu Wang <ziyu.wang@nyu.edu>.

*Proceedings of the 43rd International Conference on Machine Learning*, Seoul, South Korea. PMLR 306, 2026. Copyright 2026 by the author(s).

---

[1]Code & demo page: https://lekai-qian.github.io/BEAT-ICML2026/.

tokens within the same beat are concatenated to form a beat-level sequence. The full token sequence is then constructed by concatenating beat-level sequences in chronological order. The core idea of BEAT is to provide an efficient representation of the piano-roll format, avoiding nested or hierarchical designs that are difficult to scale (Wang et al., 2020b; Jiang et al., 2020a), while explicitly leveraging the inherent sparsity of piano-roll representations. This design ensures that each token corresponds to a fixed temporal duration of one beat, achieves compactness comparable to event-based representations, and retains the explicit temporal regularity inherent to grid-based representations.

We train an autoregressive Transformer model using our proposed tokenization scheme and compare it with existing tokenization methods across several tasks, including piano and multi-track continuation, pattern-controlled generation, and real-time music accompaniment. Both subjective and objective evaluations indicate that our method outperforms baselines on most criteria. Further analyses highlight several key advantages of our approach:

- **Compactness**: Compared to basic event- and notation-based tokenizations, BEAT produces more compact sequences. It also exhibits higher compressibility under BPE, suggesting more reusable substructures that facilitate pattern learning.

- **Long-term structural coherence**: BEAT captures long-range dependencies more effectively. The repetition–diversity analysis (Sec. 4.5, Fig. 3) and the subjective Coherence ratings both show that it achieves balanced variation and coherence across long spans, whereas existing methods tend to either introduce excessive novelty or fall into repetitive loops.

- **Real-time controllability**: BEAT provides explicit and uniform temporal representation at the beat level. This enables real-time conditioning that is difficult to achieve with event- or notation-based tokenizations.

## 2. Related Work

In this section, we review three areas of related work. We begin by comparing existing music tokenization approaches and highlighting the limitations of grid-based representations (Section 2.1). Next, we examine methods for incorporating grid-based structures into Transformer models (Section 2.2). Finally, we provide an overview of symbolic music language models and the generation tasks they enable (Section 2.3).

### 2.1. Symbolic Music Representations

**Grid-based representations**, such as piano-rolls, treat music as a 2D time-pitch matrix, analogous to images (Briot et al., 2019). This format offers direct temporal access and explicit sustain states. Piano-rolls have been adopted in hierarchical VAE models (Wang et al., 2020b), representation-learning autoencoders (Yang et al., 2019), and diffusion models for score generation (Min et al., 2023; Lv et al., 2023). Closer to our setting, MusicVAE (Roberts et al., 2018) and Measure by Measure (Yan & Duan, 2024) adopt measure-level grids, but pair them with dedicated architectures (a VAE prior and a staff-notation encoder–decoder, respectively). However, their 2D structure does not naturally fit the sequential paradigm of autoregressive Transformer language models, and severe sparsity makes direct serialization inefficient. Readers less familiar with the symbolic-music tokenization landscape may consult Appendix K for a brief background on the three representation families discussed in this section.

**Event-based representations** serialize music as sequences of MIDI control messages. Early work (Oore et al., 2020) tokenized raw MIDI events for performance modeling, and Music Transformer (Huang et al., 2019) demonstrated that Transformers with relative attention can model long-term structure over such MIDI event streams. Subsequent methods introduced explicit metrical structure: REMI (Huang & Yang, 2020) added bar and position tokens, while Compound Word (Hsiao et al., 2021) grouped attributes into compound tokens to reduce sequence length. More recently, Nested Music Transformer (Ryu et al., 2024) addresses intra-token dependencies via a sub-decoder. Despite these advances, event-based representations primarily model the temporal ordering of control actions, overlooking other inductive biases that are fundamental to music structure.

**Notation-based representations**, such as ABC notation (Walshaw, 2011) and MusicXML (Good, 2001), were originally designed for human-readable scores. Some work also explores graph-based representations, which model music as nodes connected by edges encoding temporal or harmonic relationships (Jeong et al., 2019; Karystinaios & Widmer, 2022). Recent work explores notation-based representations in LLMs: ChatMusician (Yuan et al., 2024) treats ABC as a second language for LLaMA, MuPT (Qu et al., 2024) applies BPE tokenization to ABC, and Nota-Gen (Wang et al., 2025) uses Interleaved ABC for multi-track generation. While text-compatible, notation-based methods remain less prevalent than event-based approaches. Moreover, these formats often rely on token combinations to express information, introducing complex inter-token relationships that increase sequence length and may burden sequence modeling.

### 2.2. Autoregressive Modeling of Grid-Based Data

Tokenizing grid-based data structures such as piano-rolls for Transformer input presents unique challenges. In computer

*Table 1.* Token types in BEAT encoding. The "Format" column shows the token notation used in Figure 1.

| Category | Format | Meaning | Range |
|---|---|---|---|
| Pattern | PAT$x$ | pattern token $s = x$ | 0–80 ($\tau$=4) |
| Pitch | PIT$x$ | pitch token $d = x$ | 0–127 |
| Velocity | VEL$x$ | velocity token $v = x$ | 0–127 |
| Rest | REST | empty beat marker | — |
| Instrument | INS$x$ | instrument token $i = x$ | 0–128 |
| Beat grid | BEAT, BAR | beat/bar marker | — |
| Others | TEM$x$ | tempo $= x$ | varies |
| | TS$x$ | time signature $= x$ | varies |

vision, researchers have explored various approaches to process images with Transformers. ViT (Dosovitskiy et al., 2021) partitions images into fixed-size patches and linearly embeds them as token sequences. VQ-VAE (van den Oord et al., 2017) and VQGAN (Esser et al., 2021) learn discrete codebooks to compress images into compact tokens. Building on these representations, ImageGPT (Chen et al., 2020) and LlamaGen (Sun et al., 2024) demonstrate that standard autoregressive Transformers can effectively model such discretized 2D data.

However, unlike images, piano-rolls are inherently sparse—at any given moment, only a small subset of pitches are active while the vast majority remain silent. Directly applying patch-based approaches would result in most patches being empty, while the few non-empty patches contain excessive information. Music is essentially composed of discrete events such as pitch, rhythm, and their interrelationships. The key challenge, therefore, lies in converting the grid-based piano-roll into a sparse token representation that captures these discrete musical properties.

### 2.3. Symbolic Music Generation Models

Recent years have witnessed significant progress in Transformer-based symbolic music generation. REMI (Huang & Yang, 2020) improved rhythmic coherence through beat-relative position encoding. Anticipatory Music Transformer (Thickstun et al., 2024) extends the autoregressive framework with an anticipation mechanism, enabling infilling and continuation. FIGARO (von Rütte et al., 2023) achieves fine-grained control over musical attributes such as density and instrumentation.

While these models have achieved remarkable success in offline tasks such as continuation, infilling, and conditional generation, their underlying tokenization schemes are inherently unsuitable for real-time accompaniment generation—a scenario with increasing importance. Real-time accompaniment requires the model to generate responses instantaneously as melodic input arrives. Yet, existing encoding methods typically require complete musical segments before tokenization, which precludes streaming processing. Current solutions for real-time accompaniment, such as

SongDriver (Wang et al., 2022) and RL-Duet (Jiang et al., 2020b), rely on specialized architectures or reinforcement learning strategies, and more recent work studies streaming accompaniment in the audio domain with explicit future-visibility and chunk-duration trade-offs (Wu et al., 2025). This gap between existing tokenization designs and real-time generation requirements motivates the need for a new symbolic representation that bridges this divide.

## 3. Methodology

This section introduces BEAT, our proposed grid-based tokenization framework. We begin by describing the encoding procedure (Section 3.1), followed by a discussion of its desirable structural properties (Section 3.2), and finally outline the autoregressive modeling approach (Section 3.3).

### 3.1. Beat-Wise Encoding

BEAT representation converts symbolic music with multiple tracks into a token sequence. The core idea is to treat a *beat* as the basic unit (typically aligned with a musical beat, e.g., a quarter note, though this can vary), encoding all musical events within a beat into a compact token sequence before concatenating across beats. The encoding consists of three steps: encoding each pitch's information within a beat (which we referred to as a *pattern*), assembling patterns into a beat-level sequence, and constructing the final token sequence. Figure 1 illustrates the encoding process and Table 1 summarizes the token types.

We represent symbolic music as a three-state piano-roll $X \in \{0, 1, 2\}^{P \times T}$, where $P$ is the number of pitches and $T$ is the number of time steps. Each entry $X[p, t]$ indicates the state of pitch $p$ at time $t$: 0 for silence, 1 for onset (a note attack), and 2 for sustain (a continuation of a previous onset). These $T$ time steps span $N$ beats whose boundaries follow the time signature. The parameter $\tau$ specifies the temporal resolution of a musical beat: BEAT resamples each beat's piano-roll segment to $\tau$ time steps, yielding a sequence of $N$ beat matrices $B^{(1)}, B^{(2)}, \ldots, B^{(N)} \in \{0, 1, 2\}^{P \times \tau}$.

**Step 1: Pitch pattern encoding.** Within a beat represented by its beat segment $B$, the state vector for a given pitch $p$, denoted by $\mathbf{s}_p = B[p, :] \in \{0, 1, 2\}^\tau$, describes the temporal pattern of the pitch over $\tau$ time steps. We encode this as a pattern token $s$ via base-3 conversion:

$$s_p = \sum_{t=0}^{\tau-1} \mathbf{s}_p[t] \cdot 3^{\tau-1-t}. \qquad (1)$$

In our grid-based representation, each pattern is paired with an overall velocity descriptor. In the current implementation, we use a single value $v_p$ to summarize the velocity of pitch $p$ within the beat, computed as the mean of its MIDI velocities. This provides a compact yet informative representation of

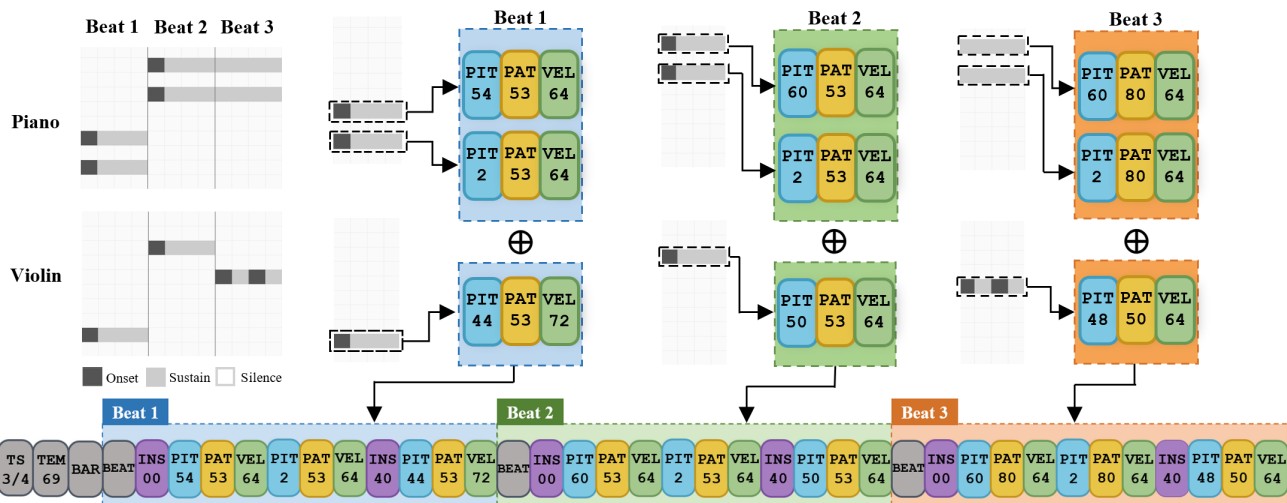

*Figure 1.* Overview of the BEAT encoding framework.

the dynamics, which can also be extended in the future.[2] We denote the resulting tokens as `PAT`$x$ for $s_p = x$ and `VEL`$x$ for $v_p = x$.

**Step 2: Beat-level assembly.** While pattern tokens capture what happens to each pitch and each beat, we need an extra token to specify positions along the pitch axis. For each beat matrix $B$, we identify the set of active pitches $\mathcal{A} = \{p : \exists\, t,\ B[p,t] \neq 0\}$. Let $M = |\mathcal{A}|$ denote the number of active pitches, sorted in descending order as $p_1 > p_2 > \cdots > p_M$. We encode pitch positions as pitch tokens $d$ using relative intervals:

$$d_1 = p_1, \quad d_j = p_{j-1} - p_j \text{ for } j \geq 2. \quad (2)$$

That is, the first pitch uses an absolute index while subsequent pitches use relative offsets. Finally, we pair each pitch token with its corresponding pattern and velocity tokens, representing the beat as:

$$\mathbf{u} = (d_1, s_{p_1}, v_{p_1}) \oplus (d_2, s_{p_2}, v_{p_2}) \oplus \cdots \oplus (d_M, s_{p_M}, v_{p_M}). \quad (3)$$

For empty beats ($M = 0$), we set $\mathbf{u} = $ `Rest`, a special token.

**Step 3: Sequence construction.** To assemble beat-level sequences $\mathbf{u}$ into a complete multi-track sequence, we introduce three additional token categories. We use *beat grid tokens* to provide temporal structure. Each beat is preceded by a `BEAT` marker that delimits beat boundaries, ensuring each beat remains a self-contained unit. `BAR` markers indicate measure boundaries.

We use *instrument tokens* to identify tracks within each beat. Each track's content is prefixed by an instrument token $i_k$, where $k$ corresponds to the MIDI program number. The

sequence for a complete piece takes the form:

$$\texttt{BEAT}\ (i_1\ \mathbf{u}_1^{(1)})\ (i_2\ \mathbf{u}_2^{(1)})\ \ \texttt{BEAT}\ (i_1\ \mathbf{u}_1^{(2)})\ (i_2\ \mathbf{u}_2^{(2)})\ \cdots, \quad (4)$$

where $\mathbf{u}_k^{(n)}$ denotes the (pitch, pattern, velocity) pairs for track $k$ at beat $n$. Parentheses are added for readability. Crucially, instrument tokens attach to beats rather than individual notes.

Finally, the sequence is augmented with *other tokens* encoding musical attributes such as tempo and time signature at bar boundaries. Mid-piece time-signature changes are handled by inserting a new `TS`$x$ token at the next measure boundary. Detailed encoding and decoding algorithms are provided in Appendix B.

### 3.2. Structural Properties of the Encoding

The BEAT tokenization exhibits several desirable structural properties:

**First, beats are explicit and localized units.** Each beat corresponds to a contiguous token subsequence, in which all musical events within that beat are grouped together. This introduces an inductive bias aligned with human perception of music in fixed time units (London, 2012), and also eliminates information leakage across beats—a benefit for conditioning in generation tasks (see Section 3.3).

**Second, the token sequence is compact and scales with musical complexity.** By leveraging the sparsity of piano-roll input, our representation scales as $O(N \cdot \bar{M})$, where $N$ is the number of beats and $\bar{M}$ is the average polyphony. This contrasts with naive piano-roll serialization and aligns with the intuition that longer or denser music requires longer descriptions.

**Third, the representation is approximately invariant under transposition and rhythm shifts.** Transposition affects

---

[2]This approximation is near-lossless on our training set; see Appendix B.2 for statistics.

only the first pitch token $d_1$; all subsequent intervals remain intact, enabling generalization across keys. Likewise, the within-beat encoding of a musical figure is invariant to its temporal position, since each beat tokenization depends solely on its local content $B^{(i)}$, not on beat index $i$. These inductive biases reduce what must be learned from data and promote generalization over musically meaningful transformations. We empirically validate how this structure enables efficient learning of given musical patterns in Section 5.2.

### 3.3. Unified Autoregressive Framework

With our BEAT tokenization, music sequences can be directly modeled using standard autoregressive Transformers: BEAT introduces no special objective or factorization, and we train with the usual next-token cross-entropy loss.

Our tokenization enables versatile control in music generation by unifying several tasks under a single autoregressive framework at the level of the token sequence, without requiring task-specific architectures. Track-conditioned generation is instantiated in Section 5.3, where accompaniment for beat $i$ is generated conditioned on the melody and the past accompaniment at beats $1, \ldots, i-1$. As with other standard tokenizations (von Rütte et al., 2023), control is achieved by conditioning on tokens for attributes such as tempo and meter, and on past tokens for continuation. A key advantage of our beat-wise encoding is its strict causal and evenly spaced structure in musical time: all information used for prediction lies strictly in the past, with no ongoing events, and each beat is represented by at least one token, ensuring no temporal skips. This makes the framework well-suited for *real-time generation*, where the model predicts the next beat (e.g., accompaniment) based on prior context (e.g., melody and past accompaniment).

## 4. Experiments

In this section, we evaluate the performance of our proposed BEAT tokenization. Comparing against baseline methods, we conduct quantitative evaluation on the task of *music continuation*, which aims to extend a 4-bar prompt in *piano* and *multi-track* formats, respectively. Section 4.1 presents the datasets used and the training details. Section 4.2 introduces the baseline tokenization methods. Our evaluation is divided into two parts: objective evaluation as detailed in Section 4.3, and subjective evaluation as covered in Section 4.4. In Section 4.5, we further examine the long-term structural coherence achieved by our method via qualitative visualizations. Additionally, ablation studies are provided in Appendix A.

### 4.1. Datasets and Training Details

We evaluate multi-track and piano continuation using different datasets. The multi-track setting is evaluated on Lakh MIDI Dataset (LMD) (Raffel, 2016), which yields 148K pieces ($\sim$8.6K hours) after processing. For the piano setting, we supplemented the 15K piano pieces found in LMD with 193K pieces from MuseScore, yielding 208K piano pieces in total. All data is quantized to 16th-note resolution and split 80/10/10 at song level with transposition augmentation.

Our language model is a 16-layer Transformer decoder following LLaMA (Touvron et al., 2023), comprising 150M parameters, with Rotary Position Embedding (RoPE) (Su et al., 2024) as the positional encoding, chosen for its length-extrapolation capability since our continuation experiments generate beyond the training context length. We refer readers to Appendix C and D for more details on data processing and model training. These settings are kept identical for our tokenization and all baseline methods to maintain a fair comparison.

### 4.2. Baseline Tokenization Methods

We compare BEAT against four representative tokenization methods. **REMI** (Huang & Yang, 2020) is a widely adopted event-based approach that serializes notes as (position, pitch, duration, velocity) events. **REMI+** (von Rütte et al., 2023) extends REMI for multi-track music by adding instrument tokens, while **Compound Word (CPW)** (Hsiao et al., 2021) aggregates the event sequence associated with each note into a single compound token. **Interleaved ABC** (Wang et al., 2025) takes a different approach, extending ABC notation to multi-track music by interleaving tracks. Additionally, we include **Anticipatory Music Transformer (AMT)** (Thickstun et al., 2024) as an external reference using their released models: AMT-Small (128M parameters, closest to our model size) and AMT-Large (780M parameters). To isolate whether BEAT's gains come from *being* grid-based or from its sparse, musically structured encoding, we additionally compare against a **Naive Piano-Roll** baseline that retains BEAT's beat-level grid but encodes each beat by enumerating all 128 MIDI pitches in a fixed order, emitting one pattern and one velocity token per pitch; full implementation details are in Appendix E.4. Implementation details of all baselines are provided in Appendix E.

### 4.3. Objective Evaluation

We introduce three metrics to evaluate music continuation: **Groove Consistency (GC)** (Dong et al., 2020), **Scale Consistency (SC)** (Dong et al., 2020), and **Fréchet Music Distance (FMD)** (Retkowski et al., 2024). **GC** measures rhythmic regularity as the similarity of onset patterns between adjacent measures. **SC** (Dong et al., 2020) measures tonal

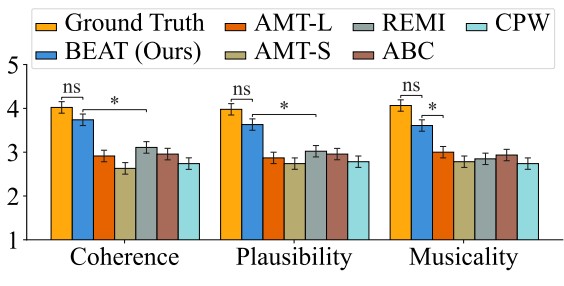

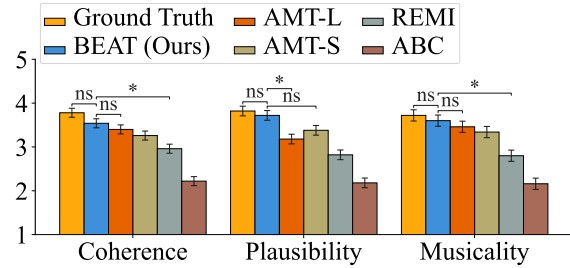

*(a) Evaluation on piano continuation.*  *(b) Evaluation on multi-track continuation*

*Figure 2.* Subjective evaluation results. Bar plots report mean ratings and standard errors. * indicates a statistically significant difference ($p < 0.05$) based on pairwise $t$-tests with Holm-Bonferroni correction; "ns" denotes non-significant differences.

*Table 2.* Objective evaluation results on music continuation (lower is better for all metrics). †Released models; all others trained with identical 150M-parameter architecture. ‡We use REMI for piano and REMI+ for multi-track; they are equivalent for single-track. "—": unsupported setting.

| | Piano | | | Multi-track | | |
|---|---|---|---|---|---|---|
| Method | $JS_{GC}\downarrow$ | $JS_{SC}\downarrow$ | FMD↓ | $JS_{GC}\downarrow$ | $JS_{SC}\downarrow$ | FMD↓ |
| Int. ABC | .677 | .023 | 522.4 | .594 | .036 | 580.0 |
| REMI(+)‡ | .552 | .038 | 550.9 | .313 | **.008** | 463.2 |
| CPW | .634 | .024 | 587.0 | — | — | — |
| AMT-S† | .603 | .055 | 447.1 | .358 | .078 | 449.3 |
| AMT-L† | .625 | .053 | 445.2 | .353 | .029 | 441.8 |
| Naive Piano-Roll | .051 | .106 | 582.3 | — | — | — |
| BEAT(Ours) | **.039** | **.021** | **436.7** | **.043** | .009 | **420.9** |

coherence as the pitch-in-scale rate. Both metrics are per-piece statistics that reflect rhythmic or harmonic regularity, and we report JS divergence between generated and ground truth distributions ($JS_{GC}$, $JS_{SC}$; lower indicates greater similarity to real music). On the other hand, **FMD** (Retkowski et al., 2024) measures the latent distributional distance between generated samples and the ground truth (lower is better), which indicates more general similarity. Detailed metric definitions are provided in Appendix F.

We evaluate piano continuation and multi-track continuation separately. For each setting, we sample 20 pieces from the test set and generate 10 continuations per prompt (truncated to 30 bars), yielding 200 samples per tokenization method. Results are presented in Table 2. REMI and Compound Word do not natively support multi-track; REMI+ extends REMI with instrument tokens. BEAT achieves the best $JS_{GC}$ and FMD on both settings, demonstrating strong rhythmic coherence and distributional similarity to real music. While some baselines achieve competitive $JS_{SC}$, they exhibit substantially worse $JS_{GC}$, indicating irregular rhythmic patterns despite reasonable tonal coherence. The Naive Piano-Roll ablation, which shares BEAT's grid but drops its sparse encoding, scores substantially worse on both $JS_{SC}$ and FMD, confirming that the gains arise from the encoding rather than the grid.

### 4.4. Subjective Evaluation

We further conduct a double-blind listening survey to evaluate music quality. Our survey contains 5 pages each for piano and multi-track continuation settings (10 pages in total). Each page begins with a 4-bar prompt drawn from the corresponding test split, followed by continuation samples generated by our method and each baseline. The ground-truth sample is also included as a perceptual anchor. Each sample is truncated to 30 bars and synthesized to audio at 120 BPM, resulting in approximately 1 minute of audio per sample. Both the page order and the sample order within each page are randomized. We request that participants listen to each sample and evaluate its musical quality on a 5-point Likert scale from 1 to 5. The evaluation considers 3 criteria: 1) *Coherence*, measuring how well the continuation fits the prompt; 2) *Plausibility*, assessing musical well-formedness and structure; and 3) *Musicality*, the overall perceptual quality. Additional details of the subjective evaluation are provided in Appendix H.

A total of 32 participants with diverse musical backgrounds completed our survey. Figure 2 shows the mean ratings and standard errors computed using within-subject ANOVA, with significant main effects (p-value $p < 0.05$) observed across all evaluation criteria. Among the baselines, REMI performs better in the piano continuation setting (Figure 2a), while AMT achieves stronger results for multi-track music (Figure 2b), which may reflect their respective design focuses. In comparison, our method consistently surpasses each baseline in both settings, yielding the highest subjective ratings overall. Post-hoc pairwise $t$-tests further reveal that, for piano continuation, our method significantly outperforms all baselines across all evaluation criteria ($p < 0.05$ with Holm-Bonferroni correction). For multi-track continuation, AMT is competitive, though our method maintains marginally higher scores. While ground-truth samples are consistently preferred over generated ones, the difference between our method and the ground truth is not statistically significant. Qualitative examples and audio samples are available in Appendix I.

## 4.5. Repetition-Diversity Analysis

To evaluate long-term structural coherence, we analyze the balance between *repetition* and *diversity* in piano continuation samples. Natural music typically exhibits thematic repetition in balance with variation. Too much repetition can make the music feel monotonous, whereas too much variation can make it feel disjointed and hard to follow.

To assess the balance level, we introduce *unique beat ratio*, a metric that quantifies the trade-off between repetition and variation. In this setting, a test piece is first converted to the piano-roll representation and segmented into beat-wise intervals. A beat interval is considered unique if it has not appeared previously. Thereby, the *unique beat ratio* at position $t$ is defined as the cumulative unique count divided by $t$. Values near 1.0 indicate high diversity, while lower values reflect increasing repetition. We use the same evaluation set as Section 4.3 (20 test pieces $\times$ 10 continuations per method, 200 samples in total), and report the average unique beat ratio across these 200 samples. We evaluate at two granularities (1-beat and 2-beat) and two matching criteria (onset-only and full state).

Figure 3 shows the unique beat ratio curves over 120 beats. BEAT closely tracks the ground truth distribution, achieving ratios within 0.3–1.2% of ground truth at beat 120, while other methods deviate by 5–30%. CPW exhibits excessive diversity (ratios 0.88–0.99 vs. ground truth 0.56–0.78), indicating a failure to develop coherent thematic material. Interleaved ABC shows excessive repetition (ratios 7–11% below ground truth). These results suggest that BEAT learns the natural repetition-variation balance present in real music.

## 5. Further Analysis

One may wonder how our model achieves superior performance despite its relatively small data scale and parameter count. In this section, we analyze the structural properties of our method and provide insights underlying its performance. Section 5.1 analyzes the tokenization compactness, which reduces training burden. Section 5.2 evaluates the ability to capture locality patterns across pitch and time, which may enhance plausibility in generation. In Section 5.3, we study an additional real-time accompaniment task, validating our method's built-in compatibility to time-aligned control.

### 5.1. Analysis of Tokenization Compactness

We investigate compactness along two dimensions: *sequence length*, and the *proportion of compressible substructures*. A tokenization that derives shorter sequences allows for modeling longer musical contexts within a fixed context window. Meanwhile, a lower compression rate suggests the presence of more regular and reusable "substrings," which may help learn more generalizable structural patterns.

Our analysis on sequence length is shown in Table 3, where BEAT produces the most compact sequences. To assess compressibility, we use Byte Pair Encoding (BPE) compression rate as a proxy. Figure 4 illustrates the compression rates under an increasing number of BPE merges. BEAT consistently achieves lower compression rates (64.83% vs. 80.15% for Interleaved ABC and 80.18% for REMI at 20 merges), indicating that BEAT fosters a higher degree of reusable substructures. This regularity may further support the recognition and generalization of structural patterns.

### 5.2. Analysis of Structural Inductive Biases

We design three pattern-constrained generative tasks as a controlled diagnostic to evaluate locality-aware structural learning, complementing the real-music evaluation in Section 4.3. For each task, we train models on synthetic data exhibiting a common pattern, using BEAT or REMI under the same architecture and hyperparameters. During generation, we measure how well the outputs capture the designated patterns. To isolate the structural advantage of beat-wise tokenization from the orthogonal effect of pitch-encoding choice, we use absolute pitch encoding for BEAT in this section (matching REMI; see Appendix A.2). This removes pitch encoding as a confound and ensures any performance gap reflects the temporal grouping structure alone. Higher pattern accuracy indicates stronger locality-aware representation learning of these structural regularities. Even though these synthetic patterns are simplifications of real music, the results suggest that BEAT more readily captures similar local regularities in actual musical sequences.

All synthetic data consist of 8-bar segments in 4/4 time. We consider three types of constrained patterns: (1) **Stepwise Transposition**, where each beat is a one-semitone transposition of the previous beat, testing whether the model learns systematic pitch-shift patterns across beats; (2) **Beat Interleaving**, where each bar follows a fixed rhythmic pattern (AAAA, ABAB, or Mixed), evaluating beat-level structural regularities; (3) **Time-Shift Reconstruction**, where the first 4 bars are repeated after a delay of 0–3 beats, testing time-invariant locality.

For Stepwise Transposition and Beat Interleaving, we evaluate pattern accuracy at both the sequence and bar levels. Results are shown in Tables 4a and 4b, where BEAT consistently outperforms REMI in both cases. For Time-Shift Reconstruction, we evaluate note-level reconstruction accuracy, where generated outputs are converted into piano-roll representations, and frame-level precision is computed for onset and sustain states, respectively. As shown in Table 4c, BEAT achieves consistently high precision (93–97%), whereas REMI struggles particularly with 2-beat shifts (78%), which corresponds to a half-bar displacement that disrupts natural bar boundaries. These results demonstrate that, compared

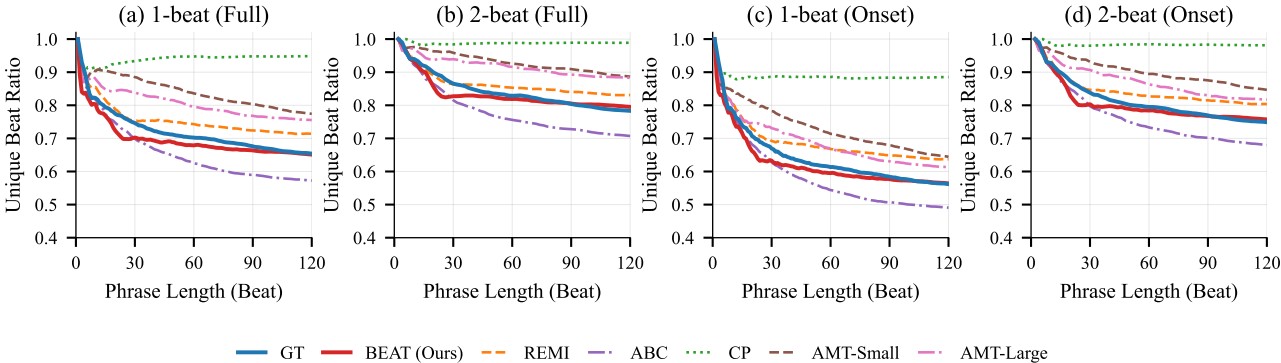

*Figure 3.* Unique beat growth curves for music continuation. The $x$-axis is beat position, i.e., the prefix length measured in beats; the $y$-axis is the cumulative unique beat ratio over the first $t$ beats, averaged across the 200 continuations per method (same set as Sec. 4.3). BEAT (red) closely tracks ground truth (blue), while Compound Word (green) shows excessive diversity and Interleaved ABC (purple) shows excessive repetition.

*Table 3.* Average token sequence length on the piano dataset. *CPW reports compound steps (1215.3 per piece), not atomic tokens.

| Method | Avg. Length |
|---|---|
| Interleaved ABC | 3450.4 |
| REMI | 1902.6 |
| CPW* | — |
| AMT | 1950.6 |
| Naive Piano-Roll | 29340.9 |
| BEAT (Ours) | **1825.6** |

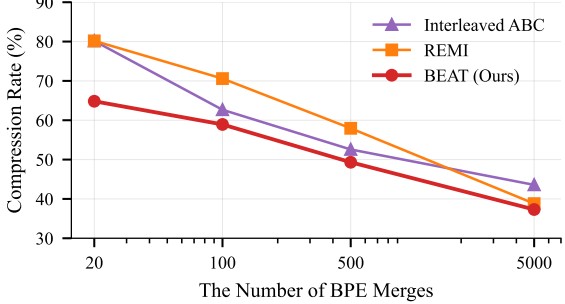

*Figure 4.* BPE compression rate across the number of BPE merges. Lower values indicate stronger regularity.

to event-based tokenization, BEAT learns stronger locality-aware representations that are robust to pitch and temporal displacement, preserving plausible and structured musical patterns. Full experimental setup for these tasks is provided in Appendix J.

### 5.3. Analysis of Real-Time Controllability

Under BEAT tokenization, a unified autoregressive framework can also be extended to real-time sequential control. In this section, we investigate the real-time accompaniment arrangement task, where the accompaniment token at time $t$ is generated based on the melody context occurring strictly

*Table 4.* Inductive bias task results comparing BEAT and REMI tokenization. Seq. = sequence-level accuracy (all 8 bars correct); Bar = per-bar accuracy. For time-shift reconstruction, we report frame-level precision on piano-roll onset and sustain states.

*(a)* Stepwise transposition accuracy (%)

| Metric | REMI | BEAT | Δ |
|---|---|---|---|
| Seq. | 28.28 | **91.92** | +63.64 |
| Bar | 84.47 | **98.48** | +14.01 |

*(b)* Beat interleaving accuracy (%)

| Pattern | Metric | REMI | BEAT | Δ |
|---|---|---|---|---|
| AAAA | Seq. | 49.48 | **82.83** | +33.35 |
| | Bar | 89.18 | **96.09** | +6.91 |
| ABAB | Seq. | 22.11 | **76.29** | +54.18 |
| | Bar | 75.66 | **93.94** | +18.28 |
| Mixed | Seq. | 7.37 | **10.31** | +2.94 |
| | Bar | 60.39 | **69.85** | +9.46 |

*(c)* Time-shift reconstruction precision (%)

| Shift | REMI | | BEAT | |
|---|---|---|---|---|
| | Onset | Sustain | Onset | Sustain |
| 0 beat | 88.3 | 89.8 | **95.5** | **96.3** |
| 1 beat | 85.4 | 89.6 | **95.0** | **97.0** |
| 2 beats | 78.0 | 79.3 | **94.2** | **96.1** |
| 3 beats | 88.7 | 89.9 | **93.9** | **95.9** |

before $t$. This setting poses challenges to event-based tokenization, as their melody and accompaniment tokens are only asynchronously aligned (Thickstun et al., 2024). In contrast, under our formulation, melody and accompaniment are treated as parallel tracks composed of beat-level synchronized units. This allows them to be interleaved unit by unit within an autoregressive framework. Specifically, the ensuing model generates accompaniment for beat $i$ conditioned on the past melody and accompaniment for beats $1, 2, \ldots, i - 1$, ensuring strict causality and fine-grained

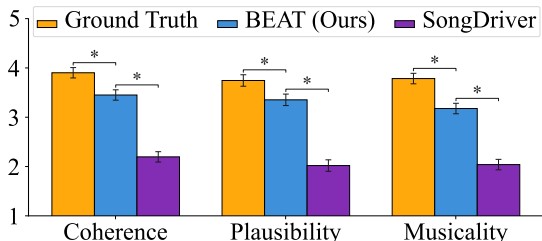

*Figure 5.* Subjective evaluation results for real-time accompaniment generation. Bar plots report mean ratings and standard errors. * indicates a statistically significant difference ($p < 0.05$).

temporal alignment.

We conduct experiment using melody-accompaniment pairs constructed from the piano test split described in Section 4.1. We fine-tune our piano continuation model on the accompaniment generation task and compare against the publicly released **SongDriver** (Wang et al., 2022), a two-stage system specifically designed for real-time accompaniment. It first generates chord labels and then selects texture candidates in a rule-based manner. In comparison, our model takes an autoregressive approach in an end-to-end fashion.

We perform a subjective evaluation in the same setup as Section 4.4. Our survey consists of 5 pages, each presenting a 30-bar melody with an initial 4-bar accompaniment prompt. It is followed by accompaniment arrangements generated by our model and SongDriver, along with the ground-truth arrangement, all in random order. Participants evaluate each sample based on 3 criteria: 1) *Coherence* (accompaniment to melody), 2) *Plausibility*, and 3) *Musicality*. As shown in Figure 5, our method significantly outperforms SongDriver, producing more coherent and plausible musical accompaniments. This highlights our method's capacity for managing fine-grained, time-aligned sequence control even under real-time causal constraints.

## 6. Limitations

A few aspects of BEAT's current scope and design choices warrant further discussion. BEAT operates on quantised input and is best suited to score-like or well-quantised MIDI; un-quantised performance MIDI, a long-standing challenge for symbolic music modelling, lies outside its current scope. Within each beat, finer temporal modelling would naturally call for a larger resolution $\tau$; however, larger $\tau$ also expands the pattern vocabulary into an increasingly long-tailed distribution that may itself hinder learning. The per-pitch velocity, currently summarised by its mean, would similarly require additional mechanisms beyond the current encoding for finer dynamic modelling. On the pitch axis, our relative pitch encoding shows a marginal advantage in large-scale generation (Appendix A.2) but its broader applicability across settings remains to be studied; we view it as a flexible design choice that can be adapted to the application.

## 7. Conclusion

To conclude, we contribute BEAT, a beat-granular tokenization for symbolic music. While retaining the compactness typically associated with event-based representations, BEAT explicitly models the temporal regularity inherent to grid-based piano-rolls, thereby preserving important musical priors such as pitch- and time-shift invariance. Owing to its uniform temporal structure, BEAT naturally supports real-time accompaniment generation, a capability that remains difficult to achieve with existing symbolic tokenizations. The subjective evaluation and the repetition–diversity analysis demonstrate that BEAT produces coherent musical content with plausible long-term structure and enhanced musical quality. We hope that BEAT's principled design will bring new perspectives to future advances in symbolic music generation and understanding.

## Impact Statement

This paper presents work aimed at advancing the field of generative music AI. Motivated by the success of large-scale language models, we propose a structured tokenization method and evaluate its generative performance against existing representations such as MIDI-like, REMI, and ABC, as a step towards building foundation models for symbolic music. Our research has several positive impacts, particularly in enhancing artistic expression and creativity. Through our experiments, we demonstrate how users can transform an initial musical idea (whether a prompt or a melody) into a complete realization. The ensuing model can assist musicians and music learners in exploring broader creative choices, thereby fostering an environment where innovation and artistic expression can thrive.

At the same time, we acknowledge the need to address potential risks. Increased accessibility to music generative models may lead to over-reliance on automation, potentially impeding the development of fundamental musical skills. We also recognize that our datasets predominantly reflect the Western musical tradition, which introduces a cultural bias that could limit the diversity of generated compositions. Widespread adoption of such models may lead to the homogenization of music, undermining the originality and individuality that are central to musical artistry.

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

## A. Ablation Study

In this section, we conduct ablation studies to investigate the effect of two key design choices: (1) temporal granularity of patterns, and (2) pitch encoding strategy. All ablation experiments are performed on the piano continuation task using the piano dataset described in Section 4.1, with identical model architecture and training setup as detailed in Appendix D.

### A.1. Temporal Granularity

In this ablation we vary the duration of the *Uniform Temporal Step* that BEAT encodes per pattern token. The default setting (used throughout the main text) is one beat per step with $\tau = 4$. Here we additionally evaluate half-beat and two-beat steps, scaling $\tau$ proportionally so that the absolute resolution within each step is held constant. We compare three settings:

- **Half-beat step** ($\tau = 2$): Each pattern token spans half a beat. Finer granularity captures more temporal detail but yields longer sequences and a smaller pattern vocabulary.

- **One-beat step** ($\tau = 4$, default): Each pattern token spans one beat. This aligns with the natural beat pulse in most Western music.

- **Two-beat step** ($\tau = 8$): Each pattern token spans two beats. Coarser granularity yields shorter sequences but may lose important rhythmic details.

Table 5 presents the results. The three settings show only modest differences in metric performance, but two practical considerations favor the one-beat configuration: the half-beat setting yields substantially longer token sequences and thus reduced modeling efficiency, while the two-beat setting enlarges the pattern vocabulary, producing a long-tailed distribution in which rare patterns receive insufficient training signal. We therefore adopt the one-beat step ($\tau = 4$) as the default throughout this paper.

*Table 5.* Ablation on the Uniform Temporal Step duration for piano continuation. $\tau$ is scaled proportionally across settings to keep the absolute resolution within each step constant.

| Temporal Granularity | $JS_{GC}\downarrow$ | $JS_{SC}\downarrow$ | FMD$\downarrow$ |
|---|---|---|---|
| Half-beat ($\tau = 2$) | 0.060 | 0.040 | 464.4 |
| One-beat ($\tau = 4$, default) | **0.039** | **0.021** | **436.7** |
| Two-beat ($\tau = 8$) | 0.145 | 0.086 | 438.2 |

### A.2. Pitch Encoding Strategy

The default BEAT encoding sorts active pitches in descending order and encodes positions using relative intervals: the first pitch uses an absolute index ($d_1 = p_1$), while subsequent pitches use relative offsets ($d_j = p_{j-1} - p_j$ for $j \geq 2$). We compare against several alternative strategies:

- **Ascending + relative**: Sort pitches in ascending order; use relative intervals.

- **Descending + absolute**: Sort pitches in descending order; use absolute pitch indices for all positions ($d_j = p_j$ for all $j$).

- **Ascending + absolute**: Sort pitches in ascending order; use absolute pitch indices.

- **Random**: Randomly order active pitches within each beat; use absolute indices (relative encoding is not applicable without a consistent ordering).

Table 6 presents the results. Relative encoding outperforms absolute on this benchmark. However, in certain scenarios we observed that absolute encoding can be more sample-efficient when training data is limited—a trade-off that warrants further study. We therefore treat relative encoding as an optional design choice rather than a strict requirement (see Section 6).

Descending and ascending orderings yield essentially the same metrics and are largely interchangeable; both, however, clearly outperform random ordering.

*Table 6.* Ablation on pitch encoding strategy for piano continuation.

| Pitch Encoding Strategy | $JS_{GC}\downarrow$ | $JS_{SC}\downarrow$ | FMD$\downarrow$ |
|---|---|---|---|
| Descending + relative (default) | **0.039** | **0.021** | **436.7** |
| Ascending + relative | 0.049 | 0.023 | 436.9 |
| Descending + absolute | 0.103 | 0.037 | 451.5 |
| Ascending + absolute | 0.119 | 0.035 | 452.6 |
| Random | 0.129 | 0.032 | 459.5 |

### A.3. Summary

The ablation study validates two key design decisions in BEAT:

1. **One-beat step ($\tau = 4$) is optimal**: This aligns with the natural beat pulse in music and provides a good balance between sequence compactness and temporal resolution.

2. **Within-beat pitch ordering should be deterministic**: a consistent ordering substantially outperforms random; the specific choice among deterministic variants (relative vs. absolute encoding, descending vs. ascending order) is a flexible design dimension.

These results support the structural properties outlined in Section 3.2 and demonstrate that BEAT's design choices are well-motivated by both musical intuition and empirical performance.

## B. Encoding and Decoding Algorithms

This section provides algorithmic details for the BEAT encoding and decoding procedures described in Section 3.1. We use consistent notation: $X \in \{0, 1, 2\}^{P \times T}$ denotes the piano-roll matrix, $\tau$ denotes steps per beat, and $N$ denotes the number of beats.

### B.1. Single-Track Encoding

Algorithm 1 describes the encoding procedure for single-track music, implementing the three steps outlined in Section 3.1: (1) partitioning the piano-roll into beat segments $B^{(i)}$, (2) encoding each active pitch as a (pitch, pattern, velocity) tuple, and (3) assembling the token sequence with beat and bar markers.

### B.2. Mean-Velocity Approximation Statistics

A potential concern about the encoding in Step 1 of Section 3.1 is that summarising the velocity of pitch $p$ within a beat by its mean may discard expressive variation. We quantified this on our training set:

(i) Across all *active* beat-pitch patterns (i.e., patterns containing at least one onset or sustain), **99.5%** contain a single distinct non-zero velocity value; in these cases the mean is identical to the original velocities and the reduction is exact. (ii) In the remaining 0.5% of patterns, the average within-pattern velocity standard deviation is **0.276** on the 0–127 MIDI velocity scale, and the mean-absolute error of replacing the per-step velocities by the pattern mean is **0.272**.

The information loss is therefore very small in practice. If finer dynamics within a beat are required (e.g., for performance-style MIDI), the framework can be extended by replacing the scalar $v_p$ with a quantized code of the full $1 \times \tau$ velocity vector, e.g., via VQ-VAE-style codebook quantization; this is a drop-in change that does not affect the rest of the encoding.

### B.3. Multi-Track Encoding

For multi-track music, we extend the single-track encoding with instrument tokens and bar-level interleaving. Algorithm 2 describes the procedure.

**Track Ordering.** Tracks are ordered by General MIDI program number in ascending order. This provides a consistent ordering across pieces: piano (0) appears before bass (32–39), which appears before strings (40–51), etc. Drum tracks (channel 10 in General MIDI) are placed last.

---

**Algorithm 1** BEAT Encoding (Single-Track)

---

**Require:** piano-roll $X \in \{0, 1, 2\}^{P \times T}$, velocity matrix $V \in \mathbb{R}^{P \times T}$, steps per beat $\tau$, beats per bar $n$
**Ensure:** Token sequence $\mathbf{E}$
1: $N \leftarrow \lceil T/\tau \rceil$ {Number of beats}
2: $\mathbf{E} \leftarrow [\,]$
3: **for** $i = 1$ to $N$ **do**
4:    **if** $(i - 1) \mod n = 0$ **then**
5:       Append BAR to $\mathbf{E}$
6:    **end if**
7:    Append BEAT to $\mathbf{E}$
8:    $B^{(i)} \leftarrow X[:, (i-1)\tau : i\tau]$ {Beat segment}
9:    $\mathcal{A} \leftarrow \{p : \exists t,\ B^{(i)}[p, t] \neq 0\}$ {Active pitches}
10:   **if** $\mathcal{A} = \emptyset$ **then**
11:      Append REST to $\mathbf{E}$ and continue
12:   **end if**
13:   Sort $\mathcal{A}$ descending: $p_1 > p_2 > \cdots > p_M$ where $M = |\mathcal{A}|$
14:   **for** $j = 1$ to $M$ **do**
15:      $\mathbf{s}_{p_j} \leftarrow B^{(i)}[p_j, :]$ {State vector}
16:      $s_{p_j} \leftarrow \text{BASE3TOINT}(\mathbf{s}_{p_j})$ {Pattern token}
17:      $v_{p_j} \leftarrow \text{MEANVELOCITY}(V[p_j, (i-1)\tau : i\tau])$ {Velocity token}
18:      **if** $j = 1$ **then**
19:         $d_1 \leftarrow p_1$ {Absolute pitch}
20:      **else**
21:         $d_j \leftarrow p_{j-1} - p_j$ {Relative interval}
22:      **end if**
23:      Append $(\text{PIT}\ d_j,\ \text{PAT}\ s_{p_j},\ \text{VEL}\ v_{p_j})$ to $\mathbf{E}$
24:   **end for**
25: **end for**
26: **return** $\mathbf{E}$

---

**Instrument Prefix.** Each track's content within a beat is prefixed by an instrument token INS $k$, where $k$ corresponds to the MIDI program number. We group similar instruments into families (e.g., all piano variants map to a single token) to reduce vocabulary size while preserving instrument identity.

**Bar-Level Interleaving.** Within each bar, tracks are serialized sequentially. This design allows the model to observe the complete musical texture at each bar before proceeding, facilitating cross-track coherence.

### B.4. Decoding

Decoding reconstructs the piano-roll matrix $X$ from a BEAT token sequence. The key insight enabling efficient decoding is that beat boundaries are implicitly marked: the first pitch token $d_1$ in each beat uses an absolute index (non-negative), while subsequent pitch tokens $d_j$ for $j \geq 2$ use relative intervals (negative). The procedure inverts Algorithm 1 accordingly and is implemented in the released code.

**Invalid Token Handling.** During autoregressive generation, the model may produce invalid tokens. We apply the following strategies:

- **Out-of-range pitch**: If the accumulated pitch $p$ falls outside $[0, P{-}1]$, skip the token and continue decoding. This preserves temporal alignment while discarding the invalid note.

- **Invalid pattern**: If $s \notin [0, 3^\tau - 1]$, treat as a rest pattern (all zeros). With $\tau = 4$, valid patterns are $[0, 80]$.

- **Unexpected token type**: If a token appears in an invalid context (e.g., BAR within a beat), skip it and continue.

These strategies ensure robust decoding even when the model produces occasional errors, maintaining temporal structure

---

**Algorithm 2** BEAT Encoding (Multi-Track)

---

**Require:** Track piano-rolls $\{X_k\}_{k=1}^{K}$, instrument programs $\{I_k\}_{k=1}^{K}$, steps per beat $\tau$, beats per bar $n$
**Ensure:** Token sequence $\mathbf{E}$
 1: Sort tracks by program number: $I_{\pi(1)} \leq I_{\pi(2)} \leq \cdots \leq I_{\pi(K)}$
 2: $N \leftarrow \lceil T/\tau \rceil, \quad N_{\text{bars}} \leftarrow \lceil N/n \rceil$
 3: $\mathbf{E} \leftarrow [\,]$
 4: **for** $m = 1$ to $N_{\text{bars}}$ **do**
 5:    Append BAR to $\mathbf{E}$
 6:    **for** $k = 1$ to $K$ **do**
 7:       Append INS $I_{\pi(k)}$ to $\mathbf{E}$
 8:       **for** $i = (m-1)n + 1$ to $\min(mn, N)$ **do**
 9:          Append BEAT to $\mathbf{E}$
10:          $\mathbf{u}_{\pi(k)}^{(i)} \leftarrow \text{ENCODEBEAT}(X_{\pi(k)}, i, \tau)$
11:          Append $\mathbf{u}_{\pi(k)}^{(i)}$ to $\mathbf{E}$
12:       **end for**
13:    **end for**
14: **end for**
15: **return** $\mathbf{E}$

---

while gracefully handling anomalies.

## C. Dataset Details

### C.1. Data Sources

Our training corpus combines two complementary sources:

- **Lakh MIDI Dataset** (Raffel, 2016): 176,581 MIDI files scraped from the web, representing a broad distribution of popular and classical music with varying quality.

- **MuseScore Collection**: approximately 1.6M user-contributed scores in MuseScore format (.mscz), crawled from the web. This collection provides higher-quality notation data with cleaner beat quantization and explicit track annotations.

The MuseScore format offers several advantages over raw MIDI: explicit beat boundaries, cleaner quantization, and richer metadata. We convert all files to a unified MIDI representation for training.

### C.2. Filtering Criteria

We filter for quality: retain pieces with 8–200 bars, remove non-standard instruments, filter quantization anomalies, deduplicate by content hash, and remove empty tracks.

### C.3. Piano Subset Construction

For accompaniment experiments, we extract pieces with exactly two piano tracks (melody and accompaniment). The melody serves as conditional input and the accompaniment as generation target.

## D. Training Details

Table 7 summarizes the model architecture used in all experiments. We adopt a decoder-only Transformer following LLaMA (Touvron et al., 2023) with Rotary Position Embedding (RoPE) (Su et al., 2024).

Table 8 lists the training hyperparameters.

**Batch Construction.** Sequences are packed to maximize GPU utilization. We concatenate multiple pieces with [EOS] tokens as separators until reaching the context length of 2048. Sequences exceeding this length are truncated.

*Table 7.* Model architecture hyperparameters.

| Hyperparameter | Value |
|---|---|
| Number of layers | 16 |
| Hidden dimension | 768 |
| Number of attention heads | 12 |
| Feed-forward dimension | 3072 |
| Dropout rate | 0.1 |
| Context length | 2048 |
| Positional encoding | RoPE |
| RoPE base frequency | 10000 |
| Total parameters | $\sim$150M |

*Table 8.* Training hyperparameters.

| Hyperparameter | Value |
|---|---|
| Optimizer | AdamW |
| Learning rate | $1 \times 10^{-4}$ |
| $\beta_1, \beta_2$ | 0.9, 0.999 |
| Weight decay | 0.01 |
| Batch size | 256 |
| Training epochs | 50 |
| LR schedule | Cosine with linear warmup (1 epoch) |
| Hardware | $4\times$ RTX 3090 |
| Training time | $\sim$400 GPU hours (50 epochs) |

**Checkpoint Selection.** We evaluate validation loss every epoch and select the checkpoint with the lowest validation loss for final evaluation.

## E. Baselines

For fair comparison, all baseline methods use identical model architecture (Section D) and training data, differing only in tokenization.

### E.1. REMI / REMI+

We implement REMI following Huang & Yang (2020). The token vocabulary includes:

- **Bar**: Bar boundary marker

- **Position**: Position within bar (0–15 for 16th-note resolution)

- **Pitch**: MIDI pitch number (0–127)

- **Duration**: Note duration in 16th notes

- **Velocity**: Quantized velocity (we use 32 bins)

REMI+ extends REMI with **Track** tokens for multi-track music, following von Rütte et al. (2023). For single-track piano, REMI+ reduces to standard REMI.

### E.2. Compound Word

We implement Compound Word following Hsiao et al. (2021). Each compound token packages multiple attributes:

$$\text{Token} = (\text{Type}, \text{Pitch}, \text{Duration}, \text{Velocity}) \tag{5}$$

The model predicts all attributes simultaneously, reducing sequence length compared to REMI. We use the same attribute vocabulary as REMI.

### E.3. Interleaved ABC

We implement multi-track ABC notation following Wang et al. (2025). Key features:

- Character-level tokenization of ABC notation

- Voice headers (V:1, V:2, etc.) for track separation

- Measure bars (—) for structural alignment

### E.4. Naive Piano-Roll Grid Baseline

This appendix details the **Naive Piano-Roll** baseline introduced in Section 4.2. It shares the beat boundary detection and the same $\tau{=}4$ resampling as BEAT, but encodes each beat differently. First, it drops the sparse encoding: for every beat, the baseline enumerates *all* 128 MIDI pitches in a fixed order ($0 \rightarrow 127$), emitting the corresponding pattern token (PAT0 for silent pitches) followed by a velocity token. The resulting per-beat block has length $2 \times 128 = 256$, regardless of polyphony. Second, it drops the relative pitch encoding: each emitted pitch token uses an absolute index PIT $p$; no descending sort or interval encoding is applied. All other settings—training data, $\tau$, the 150M-parameter LLaMA-style backbone, optimizer, batch size, training epochs—are kept identical to BEAT. This makes the Naive Piano-Roll a controlled ablation that differs from BEAT only in encoding, not in architecture or grid choice. We evaluate it on the piano continuation task (multi-track is omitted because the resulting sequence length would exceed our 2048 context window). Results are reported in Tables 2 and 3, and discussed in Section 4.3.

### E.5. Task-Specific Baselines

**Anticipatory Music Transformer** (Thickstun et al., 2024): An autoregressive model with anticipation mechanism for infilling. We use the official released checkpoint.

**SongDriver** (Wang et al., 2022): A two-stage model for real-time accompaniment. We use the official released model and follow their evaluation protocol.

Note: These baselines differ from BEAT in architecture, model size, and training data. Comparisons should be interpreted as evaluating overall system performance rather than isolated representation effects.

## F. Objective Evaluation Metrics

We adopt metrics from MusPy (Dong et al., 2020) and measure distributional similarity via Jensen-Shannon divergence.

**Groove Consistency (GC).** Groove consistency measures rhythmic regularity across measures (Dong et al., 2020). For each piece, we compute the mean similarity of onset patterns between adjacent measures:

$$\text{GC} = 1 - \frac{1}{T-1} \sum_{i=1}^{T-1} d(\mathbf{g}_i, \mathbf{g}_{i+1}) \tag{6}$$

where $T$ is the number of measures, $\mathbf{g}_i \in \{0, 1\}^R$ is the binary onset vector of measure $i$ (with $R$ time steps per measure), and $d(\cdot, \cdot)$ is the normalized Hamming distance. Higher values indicate more consistent rhythmic patterns across measures.

**Scale Consistency (SC).** Scale consistency measures tonal coherence (Dong et al., 2020). For each piece, we compute the maximum pitch-in-scale rate across the 24 standard major and natural minor scales, following MusPy's reference

implementation (harmonic and melodic minor variants are not enumerated):

$$SC = \max_{\text{root,mode}} \frac{|\{p : p \in \text{Scale}(\text{root, mode})\}|}{|\{p\}|} \tag{7}$$

where $p$ denotes pitch classes of all notes, and Scale(root, mode) defines the pitch classes belonging to the specified scale. Higher values indicate better adherence to a consistent tonal center.

**Fréchet Music Distance (FMD).** Following Retkowski et al. (2024), we compute FMD using CLaMP2 embeddings:

$$\text{FMD} = \|\mu_g - \mu_r\|^2 + \text{Tr}(\Sigma_g + \Sigma_r - 2(\Sigma_g \Sigma_r)^{1/2}) \tag{8}$$

where $(\mu_g, \Sigma_g)$ and $(\mu_r, \Sigma_r)$ are the mean and covariance of generated and reference embeddings, respectively.

## G. Out-of-Distribution Evaluation on POP909

To complement the in-distribution evaluation in Section 4.3, we additionally evaluate piano continuation on the POP909 dataset (Wang et al., 2020a), which is *not used during training* for any method. We follow the same protocol (20 test pieces $\times$ 10 continuations per method) and report FMD against POP909 ground-truth (lower is better).

*Table 9.* Out-of-distribution FMD on POP909 (lower is better). All models are trained on LMD+MuseScore and never see POP909 during training.

| Method | Int. ABC | REMI(+) | CPW | AMT-S | AMT-L | BEAT (Ours) |
|--------|----------|---------|-----|-------|-------|-------------|
| FMD↓ | 401.1 | 438.7 | 512.6 | 422.8 | 402.5 | **365.9** |

BEAT achieves the best FMD on this unseen dataset, consistent with the in-distribution results in Table 2. This indicates that the gains of BEAT over baseline tokenizations are not specific to the LMD+MuseScore training distribution.

## H. Subjective Evaluation Details

Our subjective evaluation is conducted via an online crowdsourcing study, where participants complete a survey consisting of listening and rating tasks. This section provides additional details on the survey design and participant profile.

### H.1. General Instructions

Participants receive the following general instructions, which clarify their rights and the conditions of participation:

- Participation is entirely voluntary, and one may withdraw at any time without any negative consequences.

- No personally identifying information is collected; all responses are anonymous and used solely for research purposes.

### H.2. Survey Design

Our survey consists of 5 pages for *piano continuation*, *multi-track continuation*, and *real-time accompaniment*, respectively (15 pages in total). Each page presents outputs from different models corresponding to a common test input, which are MIDI prompts or melodies drawn from the respective test split. Outputs are generated by our method and all baselines, with the ground-truth sample included as a perceptual anchor. All models to be evaluated, along with the ground truth, are anonymised, and the presentation order on each page is randomized. We distribute our survey via SurveyMonkey.[3]

Participants listen to each sample and evaluate its musical quality on a 5-point Likert scale from 1 to 5. The evaluation considers 3 criteria:

- **Coherence**: How well the continuation/accompaniment aligns with the prompt/melody.

---

[3]https://www.surveymonkey.com/

- **Plausibility**: How musically valid and well-formed the continuation/accompaniment is (considering longer-term music structure).

- **Musicality**: The overall musical quality.

To analyze the data, we first tested the normality of response distributions, and then conducted within-subject ANOVA followed by post-hoc pairwise t-tests to assess statistical significance. This ensures that observed differences in ratings reflect differences among the models rather than individual participant variability. To maintain response quality, we only included complete evaluation sets, requiring participants to rate all models on a given page for their responses to be considered valid.

### H.3. Completion Time

Each participant is randomly assigned 5 out of the 15 pages. On each page, participants first listen to the test input piece, followed by the corresponding output samples, which they then rate. All samples in the survey are 30 bars long and rendered to audio using the MuseScore[4] soundfonts, producing approximately 1min of audio per sample. This design targets a total completion time of 25 minutes, ensuring that participants have sufficient time to listen carefully without excessive fatigue. The actual completion time observed on average is 32min 25s.

### H.4. Participant Demographics and Backgrounds

Participants self-identify their musical background as *Amateur* (I enjoy listening to music; I can play, sing, or compose short pieces; I know a little music theory; I can evaluate a composition based on my feelings), *Intermediate* (I have some experience in performing, composing, or other music activities; I know a certain amount of music theory that helps me evaluate a composition), or *Professional* (I am pursuing or have completed a music degree, or have an equivalent background; I am proficient in using music theory to evaluate a composition).

Among the 32 valid responses, the demographic statistics of participants are as follows: *Age* (<18: 0%, 18–29: 62.50%, 30–44: 21.88%, 45–59: 9.38%, ≥60: 6.25%); *Gender* (Female: 18.75%, Male: 81.25%, Non-binary: 0%, Prefer not to say: 0%); *Music background* (Amateur: 37.50%, Intermediate: 31.25%, Professional: 31.25%); *Years spent on studying music* (None: 18.75%, 1 year: 6.25%, 2 years: 12.50%, 3–5 years: 12.50%, 6–10 years: 18.75%, >10 years: 31.25%). All authors are excluded from the survey.

## I. Qualitative Examples

Generated examples and interactive demonstrations are available on our demo page: `https://lekai-qian.github.io/BEAT-ICML2026/`. The source code is available at `https://github.com/Lekai-Qian/BEAT-ICML2026`.

## J. Details of Structural Inductive Bias Experiments

This section provides additional details for the structural pattern learning experiments in Section 5.2. All experiments use the same model architecture as Table 7. BEAT and event-based baselines are trained with identical hyperparameters, differing only in tokenization. BEAT uses absolute pitch encoding throughout these experiments; see Appendix A.2.

**Dataset Split.** Each task has its own set of 4,000 samples (not shared across tasks), split 9:1 into training (3,600) and validation (400). Each sample is synthesized from real 4/4-time piano data: we extract a sequence of consecutive beats from the dataset and apply the task-specific transformation to form an 8-bar segment. For example, under the AAAA pattern, a real bar with beats ABCD becomes four synthesized bars AAAA, BBBB, CCCC, DDDD (each real beat is repeated four times within a bar).

### J.1. Stepwise Transposition

**Dataset.** We construct synthetic 8-bar sequences where each beat is a one-semitone transposition of the previous beat (Figure 6). This pattern probes pitch-invariant locality.

---

[4]`https://musescore.org/`

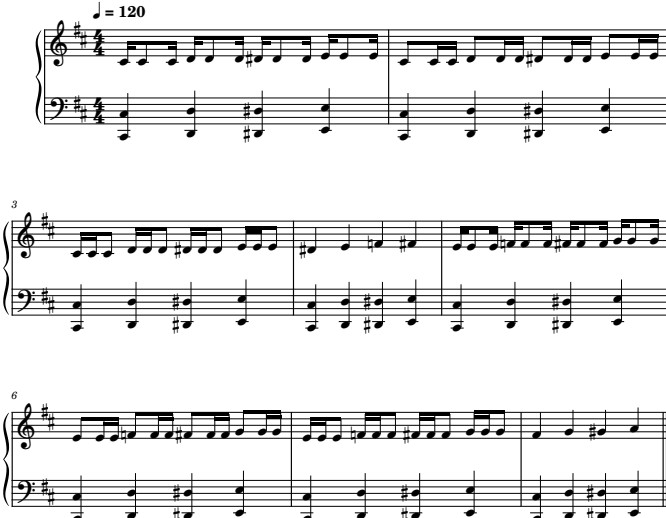

*Figure 6.* Stepwise transposition pattern: each beat is transposed up by one semitone from the previous beat.

**Evaluation.** Models generate 200 sequences unconditionally (temperature=1.0, top-p=0.95). We report:

- *Sequence Accuracy*: percentage of sequences where all 8 bars follow the target transposition pattern.

- *Bar Accuracy*: percentage of individual bars matching the expected transposition.

## J.2. Beat Interleaving

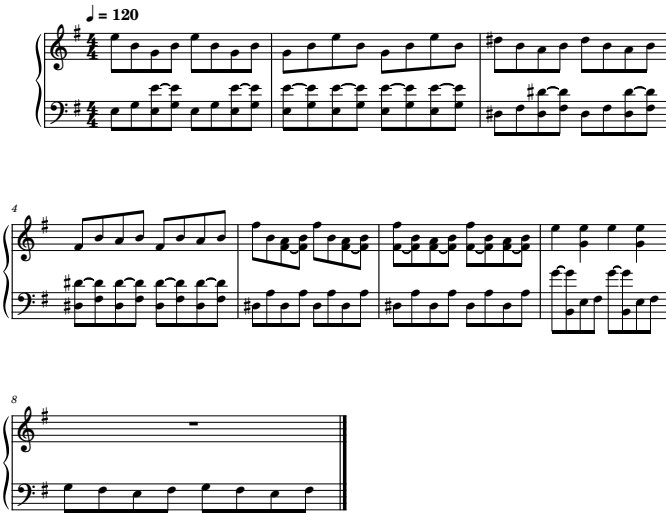

*Figure 7.* ABAB pattern: beats alternate in an A-B-A-B pattern within each bar.

**Dataset.** We construct synthetic 8-bar sequences with controlled rhythmic structures. Each bar contains 4 beats following a specific pattern:

- **AAAA** (Figure 8): Four identical beats per bar. Beat content may differ across bars while the structural pattern remains consistent.

- **ABAB** (Figure 7): Beats alternate in an A-B-A-B pattern within each bar.

- **Mixed** : AAAA and ABAB alternate across segments, testing higher-order pattern learning.

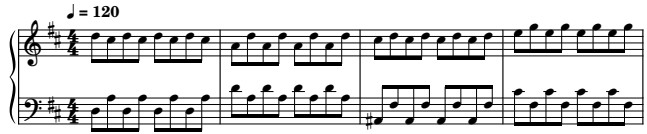

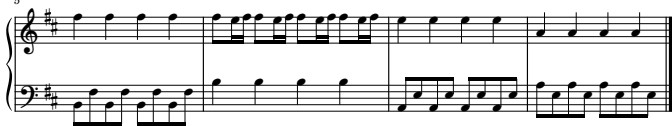

*Figure 8.* AAAA pattern: four identical beats per bar.

This task evaluates whether models can learn beat-level structural regularities.

**Evaluation.** Models generate 200 sequences unconditionally (temperature=1.0, top-p=0.95). We report Sequence Accuracy and Bar Accuracy as defined above.

### J.3. Time-Shift Reconstruction

**Dataset.** Each sequence contains a 4-bar prompt followed by the same content delayed by $k \in \{0, 1, 2, 3\}$ beats. This pattern probes time-invariant locality.

**Evaluation.** We use a held-out set of 400 samples per shift amount for final evaluation. Given the 4-bar prompt, models generate the time-shifted continuation using deterministic decoding (top-k=1). We convert outputs to piano-roll and compute frame-level precision for onset and sustain states separately, measuring note-level reconstruction fidelity.

## K. Background on Symbolic Music Tokenization

Symbolic music is commonly represented in one of three forms, each of which has given rise to its own family of deep-learning tokenizations and modeling approaches.

**MIDI control streams.** Music can be viewed as a stream of timestamped control events—note-on, note-off, time-shift, velocity, and so on. After being tokenized, these event streams map naturally onto Transformer decoders, which has made this the most actively developed family with the widest variety of tokenization designs. The most popular representative is REMI (Huang & Yang, 2020), which introduces bar and position tokens; other notable variants include Compound Word (Hsiao et al., 2021) (multi-attribute compound tokens) and the raw MIDI-like format used by Music Transformer (Huang et al., 2019). The baselines compared against BEAT in this paper are drawn primarily from this family.

**Piano-rolls.** Music can also be viewed as a two-dimensional pitch-by-time matrix—visually analogous to an image—so the community has tended to model piano-rolls with computer-vision architectures. Early GAN approaches such as MidiNet (Yang et al., 2017) and MuseGAN (Dong et al., 2018) treated piano-rolls as images, and the paradigm has since been extended to VAEs (Roberts et al., 2018; Wang et al., 2020b) and diffusion models (Min et al., 2023; Lv et al., 2023). Piano-rolls have been comparatively less explored with autoregressive language models—the gap that BEAT targets.

**Notation-based formats.** Music can be encoded as text by serializing the printed score. The most common representative is ABC (Walshaw, 2011), with recent variants such as the interleaved ABC used in NotaGen (Wang et al., 2025). Because the underlying data is already text, this family integrates naturally with large language models pre-trained on natural text (Yuan et al., 2024; Qu et al., 2024; Wang et al., 2025).

We point readers to two external resources for details on the specific tokenization variants used as baselines in this paper. For the MIDI control stream variants, we highly recommend the MidiTok library (Fradet et al., 2021) and its accompanying documentation,[5] which collects most widely-used MIDI tokenization strategies under a unified interface and provides side-by-side visual comparisons of these strategies on the same MIDI excerpt. For the notation-based variant, the specific Interleaved ABC encoding we use is detailed in NotaGen (Wang et al., 2025).

---

[5] https://miditok.readthedocs.io/en/latest/tokenizations.html

