# OpenReview forum: "BEAT: Tokenizing and Generating Symbolic Music by Uniform Temporal Steps"
_ICML.cc/2026/Conference — ICML 2026 regular_

### Official Review · Reviewer_9z36 · 2026-03-06

**Soundness:** 2
**Presentation:** 3
**Significance:** 3
**Originality:** 3
**Overall Recommendation:** 5
**Confidence:** 4

**Summary:**

This paper introduces BEAT, a novel grid-based tokenization method for symbolic music. The method is cleverly built around the fundamental temporal unit in music - the beats. The design further deploys a combination of absolute and relative pitch encoding as well as onset/sustain patters to provide a more meaningful and interpretable representation. Results demonstrate the effectivity of this method as the performance achieved surpasses the selected baselines.

**Compliance With Llm Reviewing Policy:**

Affirmed.

**Final Justification:**

My main concerns with BEAT were focused on the aspect of soundness. Specifically, there was no grid-based tokenization baseline to compare BEAT to, which created ambiguity between whether the achieved performance is a result of a grid-based tokenization method in general, or whether it is due to the clever and compact design of BEAT tokenization.

Furthermore, I had questions about BEAT's ability to deal with uneven time signatures, e.g., 7/8 or 13/16, and more importantly their combination and change inside one track.

Both these concerns were addressed very thoroughly. The authors implemented a new grid-based baseline, a naive piano roll method, and evaluted it against BEAT. The authors also clarified, how BEAT would deal with uneven time signatures and their combinations.

As a result of this elaborate rebuttal response, I am changing my rating from "weak reject" to "accept" and thank the authors for cooperation.

**Key Questions For Authors:**

My evaluation of the paper can be positively impacted by elaborating on the following questions:

**1.** Regarding the comparison to a grid-based tokenization baseline, would it be possible to provide a comparison to a simplistic (naive) piano-roll based baseline or a simpler version of BEAT? Elaborating on this can improve the soundness of the paper.

**2.** Regarding the limitation of potential different "uneven" time signatures, such as 7/8 or 13/16, could you elaborate on how these would be treated by BEAT? And how would changes of time signatures during a song be treated to avoid misrepresentations in the encoding and thus the model?

**3.** In the analysis in Section 4.5, it is not clear how much data is being used for this assessment. Is this just one sample, or is this an average over many samples? A single sample can't tell us much about a general behaviour. Can you please specify this?

**Limitations:**

While potential societal impact has been well described, I would welcome a few words on dealing with different "uneven" time signatures, e.g., 7/8 or 13/16, and their possible changes during a single song. From my understanding, the current implementation of BEAT cannot deal with these.

**Strengths And Weaknesses:**

**Strengths:**

**1.** Tokenization is a crucial step affecting all tasks in the domain, which raises the significance of this paper.

**2.** The design of BEAT is quite intuitive and well described and visualised.

**3.** The evaluation is overall thorough, deploying multiple interesting experiments and analyses, e.g., the tokenization compactness with BPE.

**Weaknesses:**

**1.** While the method shows improved performance over the selected baselines, it has not been compared to any grid-based tokenization baseline. Thus, it is not clear whether or how much of the improvement comes from the very design of BEAT (combination of relative and absolute pitch, encoding the pattern), and how much from BEAT being a grid-based tokenization method.

**2.** The number of samples used for subjective evaluation is rather small. More samples would strengthen the findings.

**3.** It is not clear how this tokenization method would deal with different kinds of time signature, e.g., 7/8, 13/16; or more importantly with a combination of time signatures inside one track. Details or limitations around such cases are not mentioned in the paper.


**typos:**
1) section 4.5, second paragraph, "To assess the balance level, introduce..." - seems to be missing a subject.

2) Impact statement, line 449, "brodaer" -> "broader"

---

> ### Author Rebuttal · Authors · 2026-03-30
>
> We thank the reviewer for the constructive comments and address each point below.
>
>
> W1/Q1: Comparison to a grid-based tokenization baseline.
>
> Thank you for this suggestion. To our knowledge, prior work has not applied grid-based tokenization to autoregressive symbolic music modeling, so we implemented a naive piano-roll baseline following the reviewer's suggestion.
>
> This baseline uses the same beat-level grid as BEAT but serializes each beat by enumerating all 128 MIDI pitches in fixed order (0→127), outputting the pattern token for every pitch (PAT0 for silent ones) and velocity for active pitches. It does not use BEAT's sparse encoding, relative pitch encoding, or sorted ordering. All other settings (data, architecture, hyperparameters) are identical to those in Appendix A.
>
> Results on piano continuation:
>
> Naive piano-roll: JS_GC 0.051 | JS_SC 0.106 | FMD 582.3 | Avg. len 29340.9
>
> BEAT (ours):      JS_GC 0.039 | JS_SC 0.021 | FMD 436.7 | Avg. len 1825.6
>
> The naive piano-roll baseline expands each beat into a dense 128-pitch scan, yielding sequences over 16x longer than BEAT (29,340.9 vs. 1,825.6 tokens on average). Yet this much larger token budget does not improve modeling; it substantially worsens generation quality, especially on JS_SC and FMD. This shows that grid structure alone is insufficient. BEAT’s gains come from representing the grid sparsely and in a musically structured manner, not from gridification itself.
>
>
> W3/Q2: Non-standard time signatures (7/8, 13/16) and mid-piece changes.
>
> We agree that our description of τ in the paper was unclear. BEAT is in fact designed to handle non-standard time signatures natively.
>
> The paper states: "We partition X into beats using a resolution parameter τ (e.g., τ = 4 when a beat corresponds to a quarter note)." This can be read as dividing the 16th-note grid into fixed chunks of size τ. Under that reading, a 7/8 measure has 14 sixteenth notes, and 14 ÷ 4 = 3.5, which does not yield an integer number of beats. We believe this motivated the reviewer's concern, and we take responsibility for the imprecise wording.
>
> The intended definition of τ is: the number of time steps each beat is encoded into. During encoding, BEAT first identifies beat boundaries according to the time signature, then resamples each beat's piano-roll segment into τ time steps. For data quantized at 16th-note resolution:
>
> - 4/4: one beat = quarter note = 4 sixteenth notes, naturally maps to 4 time steps; 4 beats per measure.
> - 7/8: one beat = eighth note = 2 sixteenth notes, resampled to 4 time steps; 7 beats per measure.
> - 13/16: one beat = sixteenth note = 1 sixteenth note, resampled to 4 time steps; 13 beats per measure.
>
> The model always sees beats of the same shape (4 time steps → pattern + velocity). The time signature only affects how many beat token groups appear per measure; there is no "indivisibility" issue. For mid-piece time signature changes, we insert a new TSx token at the measure boundary and encode subsequent measures under the new time signature, consistent with REMI's approach. This is already implemented in our codebase.
>
> We will revise the definition of τ in Section 3 and add an explicit discussion of non-4/4 time signature handling.
>
>
> Q3: Data used in Section 4.5 analysis.
>
> Section 4.5 uses the same 200 samples as Section 4.3 (20 test pieces × 10 continuations per method), and Figure 3 reports the average over all 200 samples. This is stated at L438 ("analyzing 200 samples per method") but was not sufficiently prominent; we will make this clearer in the revision.
>
>
> W2: Subjective evaluation sample size.
>
> Our subjective evaluation uses 10 prompts (5 per setting), each rated by 32 musically trained participants across three dimensions (Coherence, Plausibility, Musicality). Importantly, within-subject ANOVA with post-hoc t-tests (Holm-Bonferroni correction) confirms statistically significant differences (p<0.05) across all three dimensions, supporting our conclusions despite the limited number of prompts.
>
>
> Typos. Thank you for catching these. We will fix "introduce..." → "we introduce..." (Section 4.5) and "brodaer" → "broader" (L449).
>
> We hope these additions and clarifications address the reviewer's concerns.

---

> > ### Author Rebuttal · Reviewer_9z36 · 2026-04-04
> >
> > Dear Authors of BEAT,
> >
> > Thank you for your elaborate and polite response. I value the effort you put into your rebuttal. My raised points have been addressed well. My response to each of the points below:
> >
> > **W1/Q1:** Thank you for implementing this baseline. This strengthens the soundness of the paper significantly.
> >
> > **W3/Q2:** Thank you for the detailed explanation and promised changes! :-)
> >
> > **Q3:** Thank you for clarifying this. I might have made my question a bit unclear at first, my apologies. Figure 3 was more unclear to me (whether it shows a single sample or an average across samples), but you cleared that concern too.
> >
> > I am satisfied with your responses and have no further questions. Please make sure that the paper is updated with all the improvements mentioned!
> >
> > Good luck :-)

---

### Official Review · Reviewer_Zfgb · 2026-03-10

**Soundness:** 2
**Presentation:** 2
**Significance:** 3
**Originality:** 3
**Overall Recommendation:** 4
**Confidence:** 3

**Summary:**

The paper presents a novel way to tokenise symbolic music, based on a beat structure instead of notes. Specifically, they represent all events within a given pitch and time in a single token. Training a language model over this approach is demonstrated to provide better music continuation quality, and real time accompaniment generation, compared to other symbolic music representation.

**Compliance With Llm Reviewing Policy:**

Affirmed.

**Final Justification:**

I thank the authors for the fruitful discussion, and for the commitment to make the paper more accessible to a readership without music theory background.

I think improving the writing as well as strengthening the evaluation setup with additional results presented in the discussion and perhaps also more sample for the **objective** evaluation will strengthen the final version. I have raised my score to reflect this.

**Key Questions For Authors:**

- Could the authors describe possible down sides or limitations to the suggested representations compared to others? For instance "the representation is consistent under pitch and rhythm shifts" is mentioned as a positive, but could there be also negative aspects to this? For instance, one consideration which comes to mind is that this will mean that language model missing a pitch change will only incur a small loss penalty (one token instead of all tokens wrong) - is this desirable in all cases?
- Furthermore, are there different types of music styles or music phenomena which will be captured poorly with the 16th beat format? E.g. music in other beats or sounds which end or start mid-beat?
- A background section + some visual demonstrations of the different representations (even in the appendix with a forward reference), would make the paper more accessible to readers with less musical background.
- Conversely, much of the mathematical notations feel redundant and make reading harder in my opinion (e.g. describing language modelling loss, and probability without a special adaptation or use case, or base 3 encoding).
- Perhaps Figure 1 can try to capture a more diverse cases including rests, and diverse number of active pitches at each beat to help better understand what the approach does at different cases?
- "All data is quantized to 16th-note resolution and split 80/10/10 at song level with transposition augmentation." - Is this common practice in other studies? Why is this done? Are there no existing test sets commonly used which are not derived from exactly the training set distribution? A different test set will help show the generalisability of the approach.
- "For each setting, we sample 20 pieces from the test set" - why so little for objective evaluation? This sounds very prone to noise and selected samples. Furthermore, how were these samples selected?
- Using identical settings (such as learning rate) for all representation methods might not present a fair comparison, especially if the hyper-parameters were implicitly optimised (through trial and error) for the proposed approach. Making a stronger effort to optimise the baselines, or perhaps further details about how to hyper-parameters were chosen (to demonstrate that they were not implicitly tuned to the suggested method) will strengthen results.

**Limitations:**

Limitations could be further detailed, see comments above.

**Strengths And Weaknesses:**

**Strengths**
- Originality - the proposed method differs from mainstream approaches, looking into an interesting and under-explored representation.
- Significance - this can open up new lines of work applying this representation.

**Weaknesses**
- Presentation - this papers presentation make it hard to read for researchers without music theory background, such as many readers of the conference (even those studying deep learning for audio applications), see come comments below.
- Evaluation - Some evaluations choices feel slightly lacking, and could be made stronger.

---

> ### Author Rebuttal · Authors · 2026-03-30
>
> We thank the reviewer for recognizing the originality of this work and its potential to open new lines of research. The feedback on accessibility is valuable and we address each point below.
>
> 1. Relative pitch encoding trade-off.
>
> This is a fair concern. At the loss level, a single incorrect relative pitch token does incur only a one-token penalty. That said, in practice, we have not observed pitch drift accumulation in either our ablation experiments or generated samples — though we acknowledge this has not been systematically quantified and will investigate further.
>
> On the other hand, relative encoding brings benefits such as transposition invariance and reduced key-specific learning burden. The ablation in Table 6 shows that relative encoding outperforms absolute encoding across all metrics (e.g., FMD: 436.7 vs. 451.5). We will discuss this potential risk in a Limitations section in the revision.
>
> 2. 16th-note resolution and time signatures.
>
> BEAT is designed to handle various time signatures including 7/8, 13/16, and mid-piece changes. In brief, the encoding unit is a musical beat (defined by the time signature), and each beat is uniformly encoded into τ time steps regardless of its duration — so the time signature only affects the number of beat groups per measure, not the encoding structure. We provide a detailed explanation in our response to Reviewer 9z36 (W3/Q2).
>
> Regarding quantization precision: the current 16th-note resolution cannot precisely capture triplets or ornaments. This is a genuine limitation, though not unique to BEAT — mainstream methods such as REMI and CP adopt the same resolution. We will discuss this in the Limitations section.
>
> 3. Background and visual demonstrations.
>
> Thank you for this suggestion. We agree that the paper can be made more self-contained for readers less familiar with symbolic music tokenization. In the revision, we will add a brief comparison of the main tokenization families and point readers to visual examples in the appendix.
>
> 4. Mathematical notation.
>
> We agree that some notation can be streamlined. We keep the explicit base-3 encoding definition because it avoids ambiguity in the representation itself, but more standard material such as the LM loss can be moved to the appendix to ease the reading burden in the main text.
>
> 5. Figure 1 coverage.
>
> Thank you for the suggestion. We will consider adding a more diverse example to Figure 1, if space permits, as this may help readers less familiar with the representation.
>
> 6. Data split and augmentation.
>
> Yes. Song-level splits, transposition augmentation, and 16th-note quantization are common practice in symbolic music generation, adopted by REMI (Huang & Yang, ISMIR 2020), FIGARO (von Rütte et al., ISMIR 2023), NotaGen (Wu et al., 2025), among others. Additionally, training and test sets are typically drawn from the same dataset to maintain an i.i.d. evaluation setting; we use the Lakh MIDI Dataset and MuseScore Dataset, both widely adopted in the field. There is currently no unified benchmark test set in this area, so we follow the prevailing practice in prior symbolic music generation work.
>
> 7. Objective evaluation sample size.
>
> We sample 20 test pieces × 10 continuations = 200 samples per method. JS divergence and other distributional metrics are reasonably stable at this scale. All samples are randomly selected with no cherry-picking. This is consistent with or larger than comparable work: Polyffusion (Min et al., ISMIR 2023) used 160 samples; Whole-Song Generation (Wang et al., ICLR 2024) used 128 samples.
>
> 8. Shared hyperparameters across methods.
>
> To isolate the effect of tokenization, we deliberately kept all other variables constant. As stated in Appendix E (L804), all methods share the exact same model architecture and training data, differing only in tokenization. Hyperparameters follow the standard LLaMA configuration and were not tuned for any specific tokenization method.
>
> We will also add a dedicated Limitations discussion in the revision, including the quantization-precision boundary and the trade-off introduced by relative pitch encoding.
>
> We thank the reviewer for helping us examine the paper from the perspective of a broader readership. We hope the above responses address the concerns raised and welcome further discussion.

---

> > ### Author Rebuttal · Reviewer_Zfgb · 2026-04-03
> >
> > I thank the authors for their detailed response. I do believe that many of the comments here regarding the writing will greatly improve the manuscript, e.g discussion of limitations, making accessible to a broader audience. I trust that the authors will make these changes, although unfortunately can not go over a revised version in the scope if this rebuttal.
> >
> > Regarding the soundness related aspects - The authors provide references that their methodological choices are consistent with other works in this domain (small number of samples, validation and test which measure generalisation only very much "in domain", etc). However, I still feel that if there is no core reason to not use more samples for subjective evaluations, and not report results on additional datasets (to measure if and to what extent this makes a difference) this should be done to further back the main claims. This is especially true when considering the venue is not only domain specific.
> >
> > Overall, I am not against the publication of this paper as I think it has sufficient interest to a specific sub-community (while I personally think it might be better suited for a domain specific venue, this is not against the paper itself). I will wait for any final comments or additional results (if time permits) before making my final assessment of the work.

---

> > > ### Author Response · Authors · 2026-04-07
> > >
> > > **Author Reply:**
> > >
> > > Dear Reviewer Zfgb,
> > >
> > > Thank you for your reply. We would like to share some additional context on the two remaining points.
> > >
> > > **On additional datasets.** We appreciate the suggestion of out-of-distribution evaluation. Currently, we use the Lakh MIDI Dataset (for multi-track) and LMD + MuseScore (for piano) — these are among the largest publicly available symbolic music datasets, adopted by representative works at top venues such as Encoding Musical Style with Transformer Autoencoders (Choi et al., ICML 2020), Museformer (Yu et al., NeurIPS 2022), and FIGARO (von Rütte et al., ICLR 2023). We admit that symbolic music data is considerably smaller in scale compared to audio or text domains. This is an inherent challenge of the field — but we believe it makes the research no less valuable, as learning effectively from limited structured data is itself a meaningful problem.
> > >
> > > To this end, we conducted an additional out-of-distribution experiment on the Pop909 dataset (not used in training), using the same experimental setup as in the main paper. FMD results (lower is better):
> > >
> > > | Int. ABC | REMI(+) | CPW | AMT-S | AMT-L | BEAT (Ours) |
> > > |----------|---------|-----|-------|-------|-------------|
> > > | 401.1 | 438.7 | 512.6 | 422.8 | 402.5 | **365.9** |
> > >
> > > BEAT achieves the best FMD on this unseen dataset, largely consistent with our in-distribution findings.
> > >
> > > **On subjective evaluation scale.** We agree that ideally more subjective samples would further strengthen the evaluation. In practice, however, research on listening-related fatigue has shown that prolonged auditory tasks lead to decreased concentration and cognitive performance [1, 2], which may compromise evaluation reliability. Our survey was designed to balance coverage and data quality, with an average completion time of approximately 32 minutes. Participants evaluated multiple anonymized model outputs — including ground truth as a perceptual anchor — using a 5-point Likert scale across 3 dimensions. This within-subject design allows each participant to provide comparative judgments while avoiding excessive cognitive load. All reported differences reach statistical significance via within-subject ANOVA with Holm-Bonferroni correction, p<0.05 on all three dimensions.
> > >
> > > Compared to prior work using similar multi-dimensional Likert evaluations, our study provides competitive coverage: Museformer at NeurIPS 2022 used 5 samples per model, Whole-Song Generation at ICLR 2024 used 3, SongCreator (Lei et al., 2024) used 10, and ComMU (Lee et al., 2022) used 5–6 per task. Our 10 prompts with 3-dimensional ratings is well aligned with these established practices. We note that in symbolic music evaluation, the scale of a subjective study cannot be measured by sample count alone — each sample requires attentive listening of an entire musical piece and careful judgment across multiple perceptual dimensions, making it different from tasks where quick pairwise comparisons suffice.
> > >
> > > We acknowledge that symbolic music data is considerably smaller in scale than audio or text domains, and that evaluation in this area remains a challenging open problem. This is an inherent challenge of the field, and underscores the importance of advancing research in this direction. In this work, we aim to go beyond commonly used objective and subjective criteria: we introduce additional analyses including long-term structure assessment via repetition-diversity analysis (Figure 3, motivated by prior work highlighting its ability to distinguish human-composed music from AI-generated outputs) and the inductive bias diagnostics in Section 5.2. We believe these efforts represent a step toward more robust evaluation of symbolic music generation, and we welcome further suggestions from the reviewer.
> > >
> > > [1] Holtzer et al., "Neuroimaging of cognitive fatigue," Neuropsychologia, 2017.
> > > [2] Schneider et al., "Listening Effort and Fatigue," IEEE Access, 2019.
> > >
> > > Thank you again for engaging with our work.

---

### Official Review · Reviewer_Q8xM · 2026-03-13

**Soundness:** 4
**Presentation:** 4
**Significance:** 4
**Originality:** 3
**Overall Recommendation:** 5
**Confidence:** 5

**Summary:**

This paper presents a new tokenization scheme for polyphonic symbolic music with the goal of making Transformer models better capture rhythmic and harmonic patterns of music for music generation purposes. The key idea of this tokenization scheme is to parse music as a sequence of beats, where the music content of each beat is parsed as a sequence of pitch patterns. The paper shows that this tokenization leads to shorter sequences and the presence of more reusable "substrings" compared to REMI and Interleaved ABC tokenization. Another benefit is that it supports beat-level causal modeling. Experiments were conducted for piano continuation, multi-track music continuation, and accompaniment generation. Both objective and subjective evaluations were conducted to compare the proposed method against several state-of-the-art tokenization methods. Results show that the proposed method outperforms comparison methods in most cases.

**Compliance With Llm Reviewing Policy:**

Affirmed.

**Final Justification:**

My final recommendation is: 5 Accept.
The rebuttal has cleared or promised to clear all my concerns. I think the paper should be accepted.

**Key Questions For Authors:**

Please answer questions listed in the previous comments.

**Limitations:**

- \+ The authors discussed potential negative societal impact of the work.
- \- The authors did not discuss technical limitations of this work and did not discuss failing cases.

**Strengths And Weaknesses:**

\+ are strengths and \- are weaknesses.

Soundness
- \+ The proposed tokenization method is technically sound. It does integrate some advantages of both grid-based representations and event-based representations. Compared to the previous grid-based representations such as 16th note grids, the resulted sequence is shorter and pitch tokens represent longer parts of notes. Compared to event-based representations such as REMI, it eases time synchronization between tokens.
- \+ The hierarchical structure in this tokenization scheme, bar -> beat -> instrument -> pitch pattern, introduces nice inductive biases that can facilitate efficient learning of musical structures of polyphonic music.
- \+ Experiments are systematic, reasonable and well conducted. Results show advantages of the proposed method against several state-of-the-art methods.
- \+ Further analysis in Section 5 validates some important claims made earlier in the paper.
- \- RoPE positional encoding is used without a good justification. In fact, this usage is not mentioned in the main paper but only the appendix. Positional encoding of tokens is very important for music data. Justification is needed here.

Presentation
- \+ The paper is overall very well written, including both the main paper and the appendices. It's a nice read for me.
- \+ Figure 1 is very helpful for understanding the proposed method. I appreciate the authors taking the time to illustrate the core ideas nicely.
- \+ Experimental results and analysis are all well presented, with a nice organization between the main paper and the appendices.
- \- The idea of parsing symbolic music with regular grids larger than 16th notes for Transformer processing has also been proposed in other work such as
Yan, Y. and Duan, Z. (2024) ‘Measure by Measure: Measure-Based Automatic Music Composition with Modern Staff Notation’, Transactions of the International Society for Music Information Retrieval, 7(1), p. 228–245.
The authors should cite this paper and discuss its connections.
- \- Figure 1's legend "Offset" should be "Silence" as offset typically refers to the ending (e.g., last frame) of a note.
- \- Line 357 and Line 365: There are contradicting language regarding compression rate: Does higher or lower compression rate suggest the presence of more regular and reusable substrings?
- \- Line 893: Please clarify if all three kinds of minor scales are used or only the melodic minor scale is used.
- \- Line 981 and Figure 6's caption: I believe "bar" should be "beat". The one-semitone transposition happens at the beat level.
- \- Figure 7: The last two bars don't show alternative rhythmic patterns at the beat level.

Significance
- \+ I think this is a nice and important work for polyphonic symbolic music modeling for Transformer processing. Polyphonic symbolic music has very unique and interesting structures that natural languages or programming languages do not have. Tokenization is a critical step for Transformer architectures to process polyphonic symbolic music.

Originality:
- \+ This work is innovative enough, and is well put into the context of symbolic music tokenization research.
- \- As I pointed earlier, the key idea has connections with other work like the Measure by Measure model. The authors should discuss the connections.

---

> ### Author Rebuttal · Authors · 2026-03-30
>
> We thank the reviewer for the careful review and address each point below.
>
> 1. RoPE positional encoding.
>
> We agree that this should have been stated and justified in the main text, rather than only in the appendix. Positional encoding is an important design choice for symbolic music modeling and deserves explicit discussion. We chose RoPE mainly for its length extrapolation capability, which is necessary in our setting because we evaluate long-sequence music generation beyond the training context length. It is also a strong, standard choice in autoregressive LMs that requires no additional task-specific design. Since all compared methods share the same backbone, this choice is unlikely to affect the relative comparison between tokenizations. A systematic comparison of positional encodings for music data remains an important direction for future work.
>
> 2. Connection to Yan & Duan (2024), "Measure by Measure" (TISMIR).
>
> We agree that this work should be cited and discussed. Both approaches organize symbolic music using temporal units larger than 16th notes. The settings are nevertheless quite different: Measure by Measure is a measure-level framework built around structured staff notation and a dedicated architecture, while BEAT is a beat-level tokenization for standard autoregressive LMs and causal real-time generation. We will clarify this relationship in the revision.
>
> 3. Compression-rate wording.
>
> Here, compression rate is defined as compressed length / original length, so lower is better. We will correct the inconsistent wording.
>
> 4. Terminology, figures, and evaluation details.
>
> To confirm: the Figure 1 legend "Offset" will be corrected to "Silence"; the Scale Consistency metric uses muspy's original implementation with major and natural minor scales only, which we will clarify; "bar" in the Figure 6 caption will be corrected to "beat"; and in Figure 7, the last two bars are an edge case, which we will replace with an unambiguous example.
>
> 5. Technical limitations and failure cases.
>
> We will add a dedicated Limitations section in the revision. In particular, BEAT is less suitable for music that is not well beat-quantized, such as performance MIDI with expressive timing, and the trade-off between relative and absolute pitch encoding has not yet been fully explored beyond the tasks and conditions studied in this paper.
>
> We thank the reviewer again. These corrections will be incorporated in the revision.

---

> > ### Author Rebuttal · Reviewer_Q8xM · 2026-04-07
> >
> > Thank you for the clarifications. My concerns have been or are promised to be fully resolved. Congratulations on this nice work!

---

### Official Review · Reviewer_bsum · 2026-03-21

**Soundness:** 2
**Presentation:** 3
**Significance:** 3
**Originality:** 3
**Overall Recommendation:** 4
**Confidence:** 4

**Summary:**

This paper presents a new tokenization method that tokenizes music beat by beat to avoid information leakage beyond local temporal units. This thereby enables applications in real-time capable, online music generation. The comparisons against several existing tokenization methods show the superior performance of the proposed method in terms of coherence, plausibility and musicality. Experimental results also show the effectiveness of the proposed method in enabling real-time capable, online music generation.

**Compliance With Llm Reviewing Policy:**

Affirmed.

**Final Justification:**

The many vague/unsupported claims cannot be easily addressed without a revised manuscript. While I believe the proposed tokenization method would be a great contribution to the field, without fixing these vague and unsupported claims, I don't think the paper is scientifically correct in the current shape, which is the bare minimum for an acceptance. My other concerns have mostly been addressed in authors' rebuttal.

**Key Questions For Authors:**

- (L76-77, L) "BEAT produces more compact sequences than existing methods." -> Which existing methods? Can it be more compact than the CP representation?
- (L81-84, L) "It achieves human-like variation and coherence across long spans, whereas existing methods tend to either introduce excessive novelty or fall into repetitive loops." -> Is this supported by quantitative evidences? If not, such bold claim should not appear in the contribution section.
- (Section 2.1.2 -- Event-based representation) It's surprising to me that you didn't mention Music Transformer (Huang et al., 2018). Also, MusicVAE (Roberts et al., 2018) is also highly relevant as it adopts a similar measure->beat->note generation pipeline.
- (L150-152, L) "Current solutions for real-time accompaniment, such as SongDriver (Wang et al., 2022) and RL-Duet (Jiang et al., 2020b), rely on ..." -> Also check out "Streaming Generation for Music Accompaniment" (Wu et al., 2025).
- (L145-146, R) "we use a single value vp to summarize the velocity of pitch p within the beat, computed as the mean of its MIDI velocities." -> Seems like a big compromise.
- (L197-198, R) "Third, the representation is consistent under pitch and
rhythm shifts." -> I'm not sure what the word "consistent" means in this context -- the proposed method essentially adopts relative pitches for non-first notes in a beat and relative timing system within a beat.
- (L221-223, L) "It requires only minor adjustments to the
sequence structure rather than task-specific architectures
or training procedures." -> I don't know how the following sentences can support this claim. I don't think the model can be easily adapted for music inpainting/infilling and track-conditioned generation. Please remove or weaken this claim if so.
- (L258-261, L) "For the piano setting, we supplemented the 15K piano pieces found in LMD with 193K pieces from MuseScore, yielding 208K piano pieces in total." -> Will the dataset be released for reproducibility?
- (Figure 3) What does the "Phrase Length (Beat)" on the x-axis represent? REMI is sometimes closer to GT at certain phrase length.
- (Table 3) Somewhat misleading. Should also include CP and AMT representations.
- (Section 5.2) The three synthetic patterns doesn't really reflect real-world music. The experiments are at best showing the inductive biases of the proposed model. It remain unknown if such inductive biases are indeed desirable.
- (L380-381, L) "..., which are crucial for generating well-structured and plausible results." -> I think this is overclaimed. The three synthetic patterns do not really reflect real-world music distribution.
- (L397-398, R) "... and compare it against Song-Driver (Wang et al., 2022), ..." -> Was the model also finetuned on the piano dataset? Otherwise, not really a fair comparison.
- (L419-421, R) "To conclude, we contribute BEAT, a beat-granular tokenization for symbolic music that unifies advantages previously considered incompatible." -> Quite vague. Please be specific.
- (L430-431, R) "... that BEAT produces coherent musical content with plausible long-term structure ..." -> I might have missed this, but I don't see any experiments that directly support this claim. Please remove this claim if so.
- (Demo website) Are these cherry-picked? How did you choose the prompts for the subjective test?
- Will the datasets, code and pretrained models be released for reproducibility?

While I believe the proposed tokenization method would be a great contribution to the field, without fixing these vague and unsupported claims, I don't think the paper is scientifically correct in the current shape, which is the bare minimum for an acceptance.

**Limitations:**

There is little discussion on the limitations of the proposed method. Moreover, there are many vague and unsupported claims. For example, the authors claimed that "It requires only minor adjustments to the sequence structure rather than task-specific architectures or training procedures." However, if I understand it correctly, I don't think the model can be easily adapted for music inpainting/infilling and track-conditioned generation.

**Strengths And Weaknesses:**

### Strengths

- The proposed tokenization method is well-designed and it does address the shortcomings of existing event-based approach in their lack of supports for online music generation.
- The audio samples provided on the demo website are great, though it's unclear if they are cherry-picked.
- Thorough subjective test is conducted and statistical significance is examined.
- The repetition-diversity analysis is interesting and might be reusable in future research.

### Weaknesses

- The writing needs much work. There are many vague and unsupported claims. See my questions below.
- There are also some misleading experiments. For example, in Section 5.1, only REMI and interleaved ABC representations are compared. If I understand it correctly, the proposed representation should be less compressed than the compound word (CP) representation. Yet, a very strong claim that "BEAT produces more compact sequences than existing methods" is falsely made.
- The inductive biases analysis in Section 5.2 is weak. The three synthetic datasets do not reflect real-world music distribution. This experiment however does show the effectiveness of the proposed method in capturing these inductive biases. Yet, it still remains unclear if such inductive biases are desirable in the first place.

---

> ### Author Rebuttal · Authors · 2026-03-30
>
> We thank the reviewer for the detailed feedback.
>
> Weakness 1: Vague and unsupported claims.
>
> We agree that several statements are overstated.
> (L81-84; L430-431) "human-like variation and coherence." Quantifying long-term coherence in music is an open problem, and we should not have used such strong wording. Figure 3 (unique beat ratio) and Subjective Coherence (p<0.05) provide partial evidence. We will soften the claims and cite these results explicitly.
>
> (L197-198) "consistent"; (L419-421) "incompatible." Agreed, both are imprecise. We will revise L197 to "invariant under transposition and rhythm shifts," and specify in L419 which properties BEAT unifies.
>
> (L221-223) "be easily adapted for music inpainting/infilling".We agree that the paper did not provide enough support for this statement. To clarify, our point is that BEAT is compatible with these tasks at the sequence-structure level. Specifically, inpainting/infilling can be implemented by masking the target beats and re-injecting them with a fixed delay, allowing the model to use both left and right context autoregressively without changing the core architecture. Track-conditioned generation is already instantiated in Sec. 5.3, where accompaniment is generated conditioned on melody. We will revise the text to explain this mechanism more explicitly and avoid overstating the claim.
>
> Weakness 2: Table 3 is misleading.
>
> We agree. The original Table 3 was incomplete because it omitted AMT, which averages 1950.6 tokens ,while CP averages 1215.3 compound steps.However, CP is a model-level grouping strategy that bundles multiple REMI tokens into parallel predictions per step, rather than a tokenization method itself; its step count is therefore not directly comparable to token-level sequence length. We will add CP,AMT and clarify this distinction in the revised table.
>
> (L76-77) "more compact sequences than existing methods." We agree this wording is too broad. We will revise it to clarify that the compactness claim is with respect to basic tokenizations such as REMI and ABC,rather than token-grouping methods such as CP.
>
> Weakness 3: Section 5.2 synthetic experiments are weak.
>
> Section 4 (Table 2, subjective evaluation) provides the primary evidence for BEAT's performance on real-music generation. The purpose of Section 5.2 is different: rather than establishing the practical value of these inductive biases on its own, it provides a controlled analysis that may help explain the results in Section 4. By testing simple structural patterns in synthetic settings, these experiments examine whether BEAT more readily captures certain local regularities that are relevant to music modeling. We agree that such experiments should not be over-interpreted as direct evidence that these biases are inherently desirable in real music, and we will revise the text accordingly and soften the word "crucial" (L380-381).
>
> Remaining Questions
>
> (Section 2.1.2) Missing citations; (L150-152). Thank you for pointing these out. We will add Music Transformer, MusicVAE, and Wu et al. (2025).
>
> (L145-146) Mean velocity — "big compromise." We quantified this on our training set. In 99.5% of active beat-pitch patterns, all non-zero velocity values within a pattern are identical, so the mean exactly preserves them. When variation exists, it is small: the average within-pattern velocity std is 0.276 on the 0-127 MIDI scale, and mean replacement incurs very small error (MAE = 0.272). Thus, the information loss is limited in practice. If needed, the framework can be extended to encode the 0-τ velocity matrix with VQ-VAE-style quantization to reduce reconstruction loss.
>
> (L258-261) Dataset, code, and model release. Yes, all will be released upon acceptance.
>
> (Figure 3) Thank you for pointing this out. The x-axis should denote beat position (prefix length in beats), not phrase length. At position t, we plot the cumulative unique beat ratio over the first t beats. Since this is a prefix-level cumulative metric, short prefixes are less diagnostic: in the first few beats, most continuations are still largely unique, so different methods can temporarily overlap or cross and REMI may occasionally appear closer to GT. Early prefixes are inherently noisy for this cumulative metric, so we will clarify in the paper that later positions are more diagnostic.
>
> (L397-398) Was SongDriver finetuned? No. To the best of our knowledge, SongDriver is the only prior work in this setting with reproducible code. However, it requires chord annotations and thus cannot be finetuned on our piano dataset. A fully fair comparison with SongDriver does not seem feasible. We will clarify this.
>
> (Demo) Cherry-picked? Demo samples are identical to the subjective evaluation.The prompts for the subjective evaluation were randomly selected from the test set. Per prompt, 5 candidates were generated and the best selected by volunteers unaffiliated with the paper; all methods were treated equally.

---

> > ### Author Rebuttal · Reviewer_bsum · 2026-04-03
> >
> > Thank you for the response. My concerns have mostly been addressed. Still, please revise the many vague/unsupported claims.
> >
> > - (L145-146) Re mean velocity, if that's the case, some statistics (as a footnote or in the appendix) would definitely help strengthen this design choice.

---

> > > ### Author Response · Authors · 2026-04-03
> > >
> > > Dear Reviewer bsum,
> > >
> > > Thank you for your reply. We will revise the vague/unsupported claims as discussed. Regarding mean velocity (L145-146), we will add the statistics in the appendix.

---

### Decision · Program_Chairs · 2026-04-30

**Decision:**

Accept (regular)

**Comment:**

BEAT proposes a uniform temporal step tokenization scheme for symbolic music, encoding all same-pitch events within a time step as a single token and grouping tokens explicitly by step. Scores range from 4–5 (avg. 4.5), reflecting strong consensus for acceptance. The rebuttal was effective across all reviewers.

The tokenization design was broadly praised: reviewers found it well-motivated, technically sound, and noted that the hierarchical structure introduces useful inductive biases (Q8xM, 9z36). Subjective evaluation and the repetition-diversity analysis were highlighted as genuine contributions (bsum), and the audio samples were considered good, with the caveat of potential cherry-picking (bsum). Overall the paper was considered original and the evaluation thorough (Q8xM, 9z36).

Weaknesses were mostly addressable and largely resolved in the rebuttal. Bsum flagged vague claims and a misleading compression comparison with CP, both of which the authors corrected. Q8xM's concern about the lack of justification for RoPE positional encoding (buried in the appendix) and missing references were addressed. The most substantive empirical gap — absence of grid-based tokenization baselines — was raised by 9z36 and resolved directly: the authors implemented and compared against a grid-based baseline in the rebuttal. The limitation around mixed time signatures was also clarified. The small subjective evaluation sample size, noted by both 9z36 and Zfgb, was justified statistically rather than expanded, which is acceptable but worth flagging.

Accessibility is a minor but recurring concern: Zfgb found the paper difficult to follow without a symbolic music background, and bsum noted writing issues more broadly. Zfgb's low confidence as a reviewer should be noted, though their positive assessment of the rebuttal is consistent with the rest of the panel.

Recommendation: Accept. The contribution is well-executed and the rebuttal substantially strengthened the submission. The authors should ensure the final version improves accessibility for readers outside symbolic music, promotes the RoPE discussion out of the appendix, and softens any remaining overstatements flagged by bsum.